# Structural basis for an unprecedented enzymatic alkylation in cylindrocyclophane biosynthesis

Nathaniel R Braffman[1†], Terry B Ruskoski[2†], Katherine M Davis[3], Nathaniel R Glasser[1], Cassidy Johnson[1], C Denise Okafor[2,3], Amie K Boal[2,3]*, Emily P Balskus[1,4]*

[1]Department of Chemistry and Chemical Biology, Harvard University, Cambridge, United States; [2]Department of Biochemistry and Molecular Biology, The Pennsylvania State University, University Park, United States; [3]Department of Chemistry, The Pennsylvania State University, University Park, United States; [4]Howard Hughes Medical Institute, Harvard University, Cambridge, United States

*For correspondence:
akb20@psu.edu (AKB);
balskus@chemistry.harvard.edu
(EPB)

[†]These authors contributed equally to this work

**Abstract** The cyanobacterial enzyme CylK assembles the cylindrocyclophane natural products by performing two unusual alkylation reactions, forming new carbon–carbon bonds between aromatic rings and secondary alkyl halide substrates. This transformation is unprecedented in biology, and the structure and mechanism of CylK are unknown. Here, we report X-ray crystal structures of CylK, revealing a distinctive fusion of a $Ca^{2+}$-binding domain and a β-propeller fold. We use a mutagenic screening approach to locate CylK's active site at its domain interface, identifying two residues, Arg105 and Tyr473, that are required for catalysis. Anomalous diffraction datasets collected with bound bromide ions, a product analog, suggest that these residues interact with the alkyl halide electrophile. Additional mutagenesis and molecular dynamics simulations implicate Asp440 in activating the nucleophilic aromatic ring. Bioinformatic analysis of CylK homologs from other cyanobacteria establishes that they conserve these key catalytic amino acids, but they are likely associated with divergent reactivity and altered secondary metabolism. By gaining a molecular understanding of this unusual biosynthetic transformation, this work fills a gap in our understanding of how alkyl halides are activated and used by enzymes as biosynthetic intermediates, informing enzyme engineering, catalyst design, and natural product discovery.

## Editor's evaluation

This work has revealed an unexpected mechanism by which enzyme-catalyzed alkylation occurs. The results presented here will be broadly relevant to those pursuing enzyme engineering as well as efforts aimed toward developing small molecule inhibitors of this unusual enzyme transformation.

## Introduction

Chemists utilize alkyl halides (molecules with $sp^3$ C–X bonds, X = F, Cl, Br, or I) as key synthetic reagents because of their accessibility and favorable reactivity (**Rudolph and Lautens, 2009**); however, examples of their use as intermediates in biological systems are rare (**Adak and Moore, 2021**). A few characterized natural product biosynthetic pathways generate transiently halogenated intermediates that facilitate downstream chemical reactions in a strategy known as 'cryptic halogenation' (**Vaillancourt et al., 2005**). This biosynthetic logic requires two partner enzymes: a halogenase, which introduces the alkyl halide substituent, and a second enzyme that utilizes the halogenated intermediate

**Figure 1.** CylK and related enzymes use alkyl chloride substrates as biosynthetic intermediates. (**A**) The halogenase CylC generates an alkyl chloride substrate for CylK, which catalyzes two stepwise Friedel–Crafts alkylations to construct a paracyclophane macrocycle in cylindrocyclophane biosynthesis. This involves an intermolecular reaction between two resorcinol-containing alkyl chloride substrates (**1**) to generate intermediate (**2**), followed by an intramolecular alkylation to afford cylindrocyclophane F. (**B**) The related enzyme BrtB (30% amino acid identity, 46% similarity) catalyzes an analogous but chemically distinct C–O bond-forming event between chlorinated bartolosides and fatty acid nucleophiles.

as a substrate, leveraging its increased reactivity and releasing the corresponding halide anion as a side product. Multiple classes of halogenases have been structurally and mechanistically characterized (*Neumann et al., 2008*), but the cognate halide-utilizing enzymes in pathways that employ cryptic halogenation remain underexplored. In particular, the structures and biochemical mechanisms for enzymatically engaging and activating alkyl halide substrates as intermediates in natural product biosynthesis are unknown.

Interrogating cylindrocyclophane biosynthesis by the cyanobacterium *Cylindrospermum licheniforme* ATCC 29412 presents an exciting opportunity to study alkyl halide utilization in biological systems. In this organism, the putative diiron carboxylate halogenase CylC generates an alkyl chloride intermediate, which is further elaborated to produce resorcinol-containing alkyl chloride **1**. Two molecules of **1** are then dimerized by an alkyl chloride-utilizing enzyme, CylK, via the formation of two new carbon–carbon (C–C) bonds to construct a paracyclophane ring system (*Figure 1A*; *Nakamura et al., 2017*). CylK is annotated as a fusion of a $Ca^{2+}$-binding domain and a β-propeller fold, but the roles of these predicted protein domains in catalysis are unknown. Initial biochemical studies revealed that this transformation involves two stereoselective alkylation events that occur in a stepwise fashion, with inversion of configuration at the alkyl chloride stereocenter (*Nakamura et al., 2017*). CylK is the only enzyme known to catalyze aromatic ring alkylation with an alkyl halide electrophile, a reaction that mirrors a classical nonenzymatic reaction known as the Friedel–Crafts alkylation, which is an important transformation in organic synthesis. The traditional nonenzymatic Friedel–Crafts reaction suffers from a lack of stereo- and regiocontrol, unwanted overalkylation events, and the generation of carbocationic intermediates that can undergo unproductive rearrangements (*Friedel and Crafts, 1877*; *Roberts and Khalaf, 1984*).

In contrast, the CylK-catalyzed Friedel–Crafts alkylation overcomes these limitations, likely by tightly controlling reactivity within the enzyme active site. This enzymatic transformation holds promise for biocatalytic applications, and we have recently demonstrated CylK's ability to accept multiple discrete resorcinol nucleophiles and alkyl halide electrophiles with varying substitution and electronic character (*Schultz et al., 2019*). Moreover, uncharacterized enzymes with homology to CylK are found in

numerous cyanobacteria, one of which (BrtB) was recently revealed to catalyze carbon–oxygen (C–O) bond formation between the carboxylate groups of fatty acids and bartoloside A, an alkyl chloride-containing natural product (*Figure 1B*; *Reis et al., 2020*). This suggests that the mechanism by which CylK binds and activates alkyl chlorides might be shared with this and other homologs that use diverse nucleophilic substrates.

Despite the intriguing reactivity and potential applications of CylK and related alkyl halide activating enzymes, our knowledge of its structure and mechanism is limited. In this work, we present a crystal structure of CylK, identify critical active site residues, provide experimental and bioinformatic support for their roles in catalysis, and propose a mechanism by which alkyl chloride substrates are activated for stereospecific nucleophilic substitution. In determining how CylK performs this Friedel–Crafts alkylation, we have enhanced our fundamental understanding of how enzymes engage alkyl halide substrates. Our structural information and mechanistic model will guide future enzyme engineering efforts, inform the design of nonenzymatic catalysts, and enable genome mining to uncover new natural products constructed by related biosynthetic strategies.

## Results
### CylK is a distinctive fusion of two protein domains

CylK crystallized in the $C222_1$ space group with a single monomer in the asymmetric unit (*Table 1*). Suitable molecular replacement (MR) models for phasing could not be identified. Phase information was obtained by partial substitution of native $Ca^{2+}$-binding sites in CylK with $Tb^{3+}$, followed by collection of X-ray diffraction datasets at the Tb X-ray absorption peak energy. The structure of CylK was solved to 1.68 Å resolution via a combined MR and single-wavelength anomalous diffraction (SAD) approach, with the MR search model generated from a poly-alanine seven-bladed β-propeller. The structure reveals an unprecedented fusion of two protein folds, an N-terminal $Ca^{2+}$-binding domain and a C-terminal β-propeller domain (*Figure 2A*).

The N-terminal domain of CylK contains a short helical component that packs against a β-roll core (*Figure 2B*, *Figure 2—figure supplement 1*). The core fold is structurally similar to repeat-in-toxin (RTX) motifs, found in diverse bacterial extracellular proteins (*Linhartová et al., 2010*). A search of the PDB for structural relatives of the CylK N-terminal domain reveals similarity to the C-terminal RTX domains of secreted toxins (*Motlova et al., 2020*), such as the adenylate cyclase toxin of *Bordetella pertussis*, and bacterial surface-layer proteins (*von Kügelgen et al., 2020*), such as the RsaA protein of *Caulobacter crescentus* (*Table 2*). In these systems, the RTX domain facilitates $Ca^{2+}$-dependent folding or assembly in the calcium-rich environment outside the cell but not in the calcium-depleted cytosol. The CylK N-terminal domain also resembles RTX domains fused to catalytic domains of extracellular hydrolases and epimerases from Gram-negative bacteria (*Figure 2—figure supplement 2*; *Tanaka et al., 2007*; *Buchinger et al., 2014*; *Meier et al., 2007*). In these systems, the RTX unit is attached to the C-terminus of the catalytic domain to facilitate extracellular secretion and $Ca^{2+}$-dependent folding. The RTX component of these characterized proteins is functionally modular, with no obvious direct connection to the active site, although in some systems that target large biopolymer substrates, the RTX domain may help anchor the substrate via electrostatic interactions (*Buchinger et al., 2014*).

The N-terminal domain of CylK differs from typical RTX motifs in several important ways. Canonical RTX motifs have an extended compact oval structure shaped by tight turns between subsequent β-strands that are stabilized by stacked metal-binding motifs on both sides of the core fold (*Linhartová et al., 2010*). The metal-binding sites are formed by a repeating GXXGXD motif in the tight turn. Vertically stacked pairs of these repeats yield hexacoordinate $Ca^{2+}$-binding sites with four carbonyl ligands and two carboxylates provided by the conserved Asp side chains. CylK adopts a more asymmetric version of this fold and contains just a single copy of the consensus $Ca^{2+}$-binding motif. The modified β-roll structure in CylK contains three vertically aligned $Ca^{2+}$ ions on one side of the core fold. The other side lacks the metal-binding sites found in other RTX proteins. Additionally, while the first two turns of the β-roll in CylK form tight junctions near the $Ca^{2+}$ ions, all of the subsequent turns contain long extensions (*Figure 2—figure supplement 1*), some of which facilitate integration with the C-terminal domain. Interestingly, the section of the β-roll motif that conserves the $Ca^{2+}$ sites packs closely into the center of the C-terminal domain. Although the CylK N-terminal domain is

**Table 1.** Data collection and refinement statistics for Tb-soaked *C. licheniforme* CylK structures.

| | Tb anomalous | Native (final model) |
|---|---|---|
| **Data collection** | | |
| Space group | *C 2 2 2₁* | *C 2 2 2₁* |
| Wavelength (Å) | 1.6314 | 0.97872 |
| Cell dimensions | | |
| *a, b, c* (Å) | 94.265, 139.286, 100.381 | 93.667, 138.613, 99.438 |
| α, β, γ (°) | 90.0, 90.0, 90.0 | 90.0, 90.0, 90.0 |
| Resolution (Å) | 50–2.38 (2.42–2.38) | 50.0–1.68 (1.71–1.68) |
| $R_{merge}$ | 0.218 (0.670) | 0.101 (1.326) |
| $R_{pim}$ | 0.041 (0.192) | 0.026 (0.346) |
| $I / \sigma I$ | 25.2 (3.8) | 29.3 (2.15) |
| $CC_{1/2}$ | 1.010 (0.906) | 0.999 (0.780) |
| Completeness (%) | 99.5 (99.9) | 99.8 (99.3) |
| Redundancy | 29.5 (11.4) | 14.8 (14.3) |
| **Refinement** | | |
| Resolution (Å) | | 42.41–1.68 |
| No. reflections | | 68,476 |
| $R_{work}/R_{free}$ | | 0.1556/0.1894 |
| No. atoms | | 5,292 |
| Protein | | 4,903 |
| Ligand/ion | | 50 |
| Water | | 339 |
| *B*-factors (Å²) | | |
| Protein | | 18.20 |
| Ligand/ion | | 28.54 |
| Water | | 25.82 |
| RMS deviations | | |
| Bond lengths (Å) | | 0.016 |
| Bond angles (°) | | 1.921 |
| Molprobity clashscore | | 2.29 (99th percentile) |
| Rotamer outliers (%) | | 0.78 |
| Ramachandran favored (%) | | 96.33 |
| Ramachandran allowed (%) | | 3.51 |
| Ramachandran outlier *(%) | | 0.16 |
| PDB accession code | | 7RON |

*Values in parentheses are for highest-resolution shell.

topologically distinct from other β-roll motifs, CylK shares with these systems a functional requirement for $Ca^{2+}$ (***Nakamura et al., 2017***).

The C-terminal domain of CylK adopts a seven-bladed β-propeller fold (***Figure 2C***, ***Figure 2— figure supplement 3***), a widespread structural motif found in both eukaryotic and prokaryotic protein

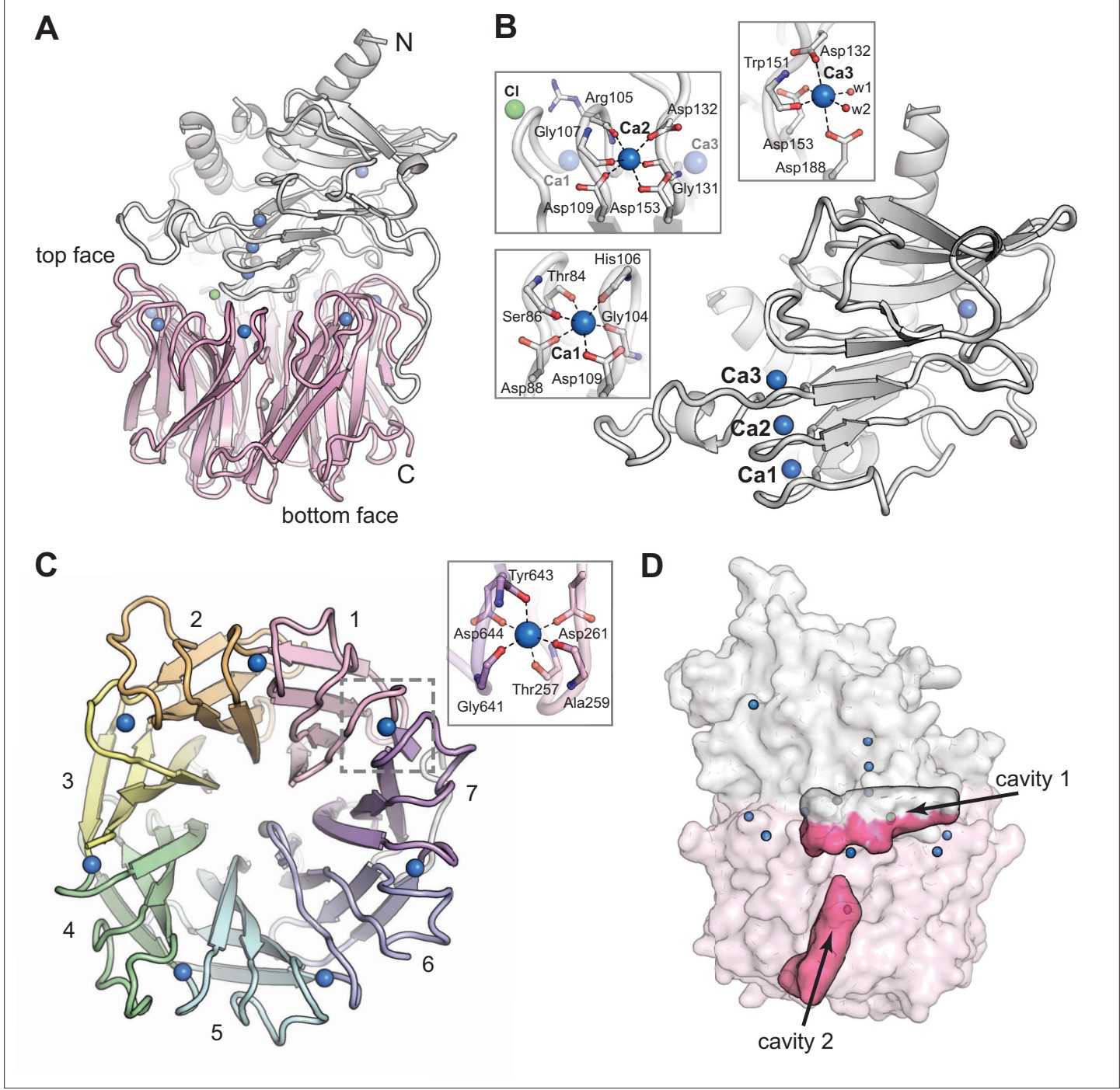

**Figure 2.** The CylK X-ray crystal structure reveals a distinct arrangement of two protein domains. (**A**) An overall view of the structure of CylK. In this image and throughout, calcium ions are shown as blue spheres, magnesium ions as dark gray spheres, and chloride ions as green spheres. The N-terminal domain is depicted as a light gray ribbon diagram, and the C-terminal domain is shown as a pink ribbon diagram. (**B**) The N-terminal domain contains a right-handed parallel β-roll stabilized by three $Ca^{2+}$ ions. The structure is capped by a three-strand antiparallel β-sheet and buttressed by additional helical secondary structures. Insets show the $Ca^{2+}$ ion coordination environment within the β-roll. (**C**) A top-down view of the seven-bladed β-propeller C-terminal domain. The propeller blades are numbered from the N-terminal end of the domain and colored by blade. The inset shows a representative view of the $Ca^{2+}$ coordination environment. A similar binding site exists at each blade junction. (**D**) Cavity mapping analysis (***Ho and Gruswitz, 2008***) of CylK reveals two cavities large enough to accommodate the alkyl resorcinol substrates.

The online version of this article includes the following figure supplement(s) for figure 2:

**Figure supplement 1.** Topology diagram of the N-terminal CylK β-roll domain, residues 7–251.

*Figure 2 continued on next page*

*Figure 2 continued*

**Figure supplement 2.** Comparison of the N-terminal CylK β-roll domain to similar C-terminal repeat-in-toxin (RTX) domains in other enzymes.

**Figure supplement 3.** Topological and structural features of the C-terminal CylK β-propeller domain.

**Figure supplement 4.** Comparison of the proposed CylK active site location to those of other characterized seven-bladed β-propeller enzymes.

**Figure supplement 5.** Solvent accessibility and properties of the cavity located in between the C-terminal β-propeller domain and the N-terminal domain in CylK.

**Figure supplement 6.** A comparison of the overall structures of CylK and selected seven-bladed β-propeller enzymes.

structures (*Chen et al., 2011*). In eukaryotes, this flat, cylindrically shaped fold facilitates protein–protein interactions (*Schapira et al., 2017*) via the same interface (top face) that interacts with the N-terminal domain in CylK. In prokaryotes and some eukaryotes (plants), β-propeller folds can be used in catalysis (*Chen et al., 2011*). Comparison of the CylK β-propeller domain to other examples of this fold in bacterial enzymes reveals several structural differences unique to CylK (*Figure 2—figure supplement 4*). In all β-propellers, adjacent four-stranded 'blades' are connected back-to-front by loops on the top face of the domain. The top face of the fold also consists of loops that link the internal β-strands of each propeller blade. In CylK, these internal loops are unusually long and folded back over the outside of the central propeller fold. Also, the loops that connect subsequent blades contain unique blade-bridging $Ca^{2+}$-binding sites, suggesting a shared role for this divalent metal in structural stabilization of both domains of CylK.

CylK is structurally similar to a class of single-domain bacterial β-propeller enzymes implicated in streptogramin antibiotic resistance that includes virginiamycin B lyase (Vgb) (*Table 3*; *Korczynska et al., 2007*; *Lipka et al., 2008*). These enzymes share with CylK an ability to interface with macrocyclic substrates or products, but they perform a distinct chemical transformation. While CylK forms two C–C bonds to generate a cyclic product, the streptogramin resistance proteins linearize cyclic peptide substrates by cleaving a C–O bond. X-ray crystal structures of an inactive variant of Vgb bound to a substrate analog provide insight into the location of the active site and mechanism of macrocycle opening (*Korczynska et al., 2007*). In Vgb, the substrate analog and an $Mg^{2+}$ ion necessary for catalysis bind near conserved polar side chains in blades 6 and 7 of the propeller fold (*Figure 2—figure supplement 4B*). In CylK, these sites are substituted for hydrophobic side chains, suggesting that, despite the similarities to streptogramin lyases, the substrates of CylK likely bind in a different location and are transformed by a distinct mechanism.

## The CylK active site is located at the domain interface

To identify CylK's active site, we focused on two solvent-accessible cavities (*Figure 2D*). Cavity 1 is lined by the top face of the β-propeller domain and the N-terminal domain, forming a solvent-accessible tunnel that is ~18 Å deep and 15 Å wide at its largest point. The walls of this cavity contain both charged/polar residues and hydrophobic patches, ideal for interaction with the amphipathic resorcinol substrates (*Figure 2—figure supplement 5*). We also interrogated a second central channel located exclusively within the C-terminal domain and opening to the opposite (bottom face) side of the propeller motif (cavity 2). To identify the active site, we individually mutagenized 17 polar residues spanning the two cavities (*Figure 3A*) that are conserved in putative CylK homologs from cyanobacteria known to produce structurally related natural products (*Figure 3—figure supplement*

**Table 2.** Selected structural homologs of the N-terminal domain of CylK obtained by a structural comparison to other proteins in the PDB (*Gáspári, 2020*).

| Rank | PDB ID | Functional description | Z-score | RMSD | No. residues aligned | Total residues |
|---|---|---|---|---|---|---|
| 1 | 1OMJ-A | Psychrophilic alkaline metalloprotease (PAP); serralysin family | 11.2 | 6.8 | 135 | 456 |
| 2 | 2Z8Z-A | MIS38 lipase | 11.1 | 8.5 | 116 | 616 |
| 3 | 6SUS-A | RTX domain of blocks IV and V of adenylate cyclase toxin | 11.0 | 3.1 | 120 | 258 |
| 4 | 2QUA-A | LipA, extracellular lipase from *Serratia marcescens* | 10.6 | 2.3 | 110 | 615 |
| 5 | 2ML1-A | AlgE6R1, Mannuronan C5-epimerase | 10.6 | 3.0 | 122 | 153 |

**Table 3.** Selected structural homologs of the C-terminal domain of CylK obtained by a structural comparison to other proteins in the PDB (*Gáspári, 2020*).

| Rank | PDB ID | Functional description | Z-score | RMSD | No. residues aligned | Total residues |
|---|---|---|---|---|---|---|
| 1 | 1SQ9-A | Antiviral protein Ski8 | 31.2 | 2.4 | 286 | 378 |
| 2 | 4H5J-B | Guanine nucleotide-exchange factor Sec12 | 31.1 | 2.9 | 300 | 347 |
| 3 | 5TF2-A | Prolactin regulatory element-binding protein | 31.0 | 2.7 | 291 | 338 |
| 4 | 4J0X-B | Ribosomal RNA-processing protein 9 | 30.9 | 2.4 | 290 | 366 |
| 5 | 2Z2O-C | Virginiamycin B lyase | 30.5 | 2.4 | 286 | 299 |

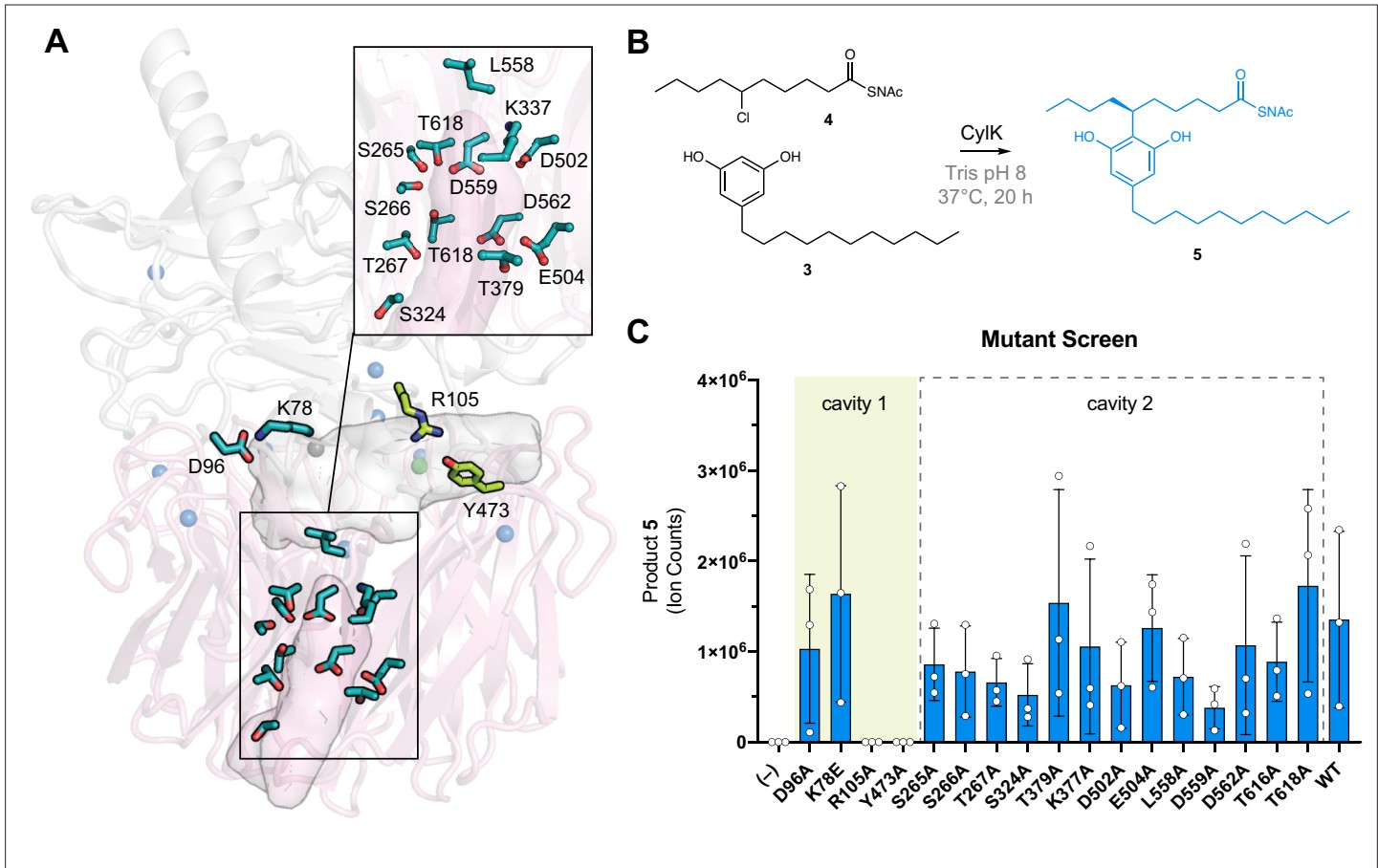

**Figure 3.** The active site of CylK is located at the domain interface. (**A**) Selected amino acids in CylK near cavities 1 and 2 subjected to mutant scanning are shown in stick format. Side chains shown in yellow near cavity 1 correspond to sites found to be essential for activity. (**B**) Non-native substrate pair used in CylK activity assays and product of the alkylation reaction (SNAc = *N*-acetylcysteamine). (**C**) Screen of mutant activity to locate the CylK active site. Product formation was measured by liquid chromatography-mass spectrometry (LC-MS), error bars represent the standard deviation from the mean of three biological replicates.

The online version of this article includes the following figure supplement(s) for figure 3:

**Figure supplement 1.** Structurally related, cylindrocyclophane-like natural products produced by cyanobacteria with related biosynthetic gene clusters and CylK homologs, namely, merocyclophanes (*Nostoc* sp. UIC 10110, MerH), cylindrofridins (*Cylindrospermum stagnale* PCC 7417, CylK), and carbamidocyclophanes (*Nostoc* sp. CAVN2, CabK).

**Table 4.** Conservation of CylK residues selected for mutagenesis experiments.

| Position | Residue (CylK, *C. licheniforme*) | Mutant | Residue (CabK) | Residue (CylK, *C. stagnale*) | Residue (MerH) | Residue (BrtB) |
|---|---|---|---|---|---|---|
| 78 | Lys | Glu | *Lys* | *Lys* | *Lys* | *Lys* |
| 96 | Asp | Ala | *Asp* | *Asp* | *Asp* | *Asp* |
| 105 | Arg | Ala, Lys | *Arg* | *Arg* | *Arg* | *Arg* |
| 473 | Tyr | Ala, | *Tyr* | *Tyr* | *Tyr* | *Tyr* |
| 265 | Ser | Ala | *Ser* | *Ser* | Ala | Ala |
| 266 | Ser | Ala | *Ser* | *Ser* | Ser | Thr |
| 267 | Thr | Ala | *Thr* | *Thr* | *Thr* | Ala |
| 324 | Ser | Ala | *Ser* | *Ser* | Ser | *Ser* |
| 377 | Lys | Ala | Arg | *Lys* | Arg | Gly |
| 379 | Thr | Ala | *Thr* | *Thr* | *Thr* | Ala |
| 502 | Asp | Ala | *Asp* | *Asp* | Asp | Asp |
| 504 | Glu | Ala | *Glu* | *Glu* | *Glu* | Ala |
| 558 | Leu | Ala | *Leu* | Leu | *Leu* | Met |
| 559 | Asp | Ala | *Asp* | *Asp* | Asp | Gly |
| 562 | Asp | Ala | His | *Asp* | Glu | Val |
| 616 | Thr | Ala | *Thr* | *Thr* | *Thr* | Gly |
| 618 | Thr | Ala | *Thr* | *Thr* | Ser | Ser |
| 374 | Glu | Ala | *Glu* | *Glu* | *Glu* | Arg |
| 438 | Leu | Ala | *Leu* | *Leu* | *Leu* | *Leu* |
| 440 | Asp | Ala, Asn | *Asp* | *Asp* | Asp | Glu |
| 499 | Phe | Ala | *Phe* | *Phe* | *Phe* | Tyr |
| 318 | Ser | Ala | *Ser* | *Ser* | Thr | Ile |
| 334 | Asn | Ala | *Asn* | *Asn* | Asn | Trp |

*1*, *Table 4*; *Preisitsch et al., 2015*; *May et al., 2017*; *Preisitsch et al., 2016*; *Leão et al., 2015*). To rapidly test variant proteins for activity, we designed a plate-based lysate activity assay using non-native substrate pair **3 + 4** (*Figure 3B*), which was previously demonstrated to be accepted by CylK (*Schultz et al., 2019*). All variants associated with the central/bottom channel of the C-terminal β-propeller retained full alkylation activity (*Figure 3C*). Strikingly, substitution of two residues located at the domain interface between the top face of the β-propeller and the N-terminal domain, Arg105 and Tyr473, completely abolished all activity on substrate pair **3 + 4** (*Figure 3C*). Notably, although located in close proximity, Arg105 is supplied by the N-terminal domain, while Tyr473 is from the C-terminal β-propeller, suggesting that both domains play an essential role in catalysis. These data preliminarily supported the location of the active site as the cleft formed between the two domains of CylK (cavity 1).

In parallel, we hypothesized that the alkyl chloride moiety may be an important substrate-binding determinant based on our inability to obtain a full occupancy complex of resorcinol **3** in CylK crystals. Although **3** was required for CylK crystallization, we could not model the alkyl resorcinol substrate in the resulting structure. To visualize locations within CylK capable of binding an alkyl halide or free halide, we soaked CylK crystals with NaBr solutions. The Br⁻ ions are surrogates for the native Cl⁻ side product and are generated in reactions with alkyl bromide electrophiles, which CylK was previously shown to accept with similar efficiency to alkyl chlorides (*Schultz et al., 2019*). Bromine X-ray absorption energies can be easily accessed at a conventional synchrotron X-ray source for anomalous

diffraction experiments, allowing us to pinpoint the location of the halide in the structure. Initial refinements identified two strong positive peaks located ~15 Å apart in $F_o–F_c$ electron density maps of the interdomain cavity (cavity 1). Anomalous diffraction datasets collected at the absorption edge of bromine confirmed assignment of these peaks as Br⁻ ions (*Figure 4A*, *Table 5*). Anomalous peaks were not found in the central channel of the β-propeller domain (cavity 2).

The bromide peak of highest intensity (~33 σ) in the anomalous difference electron density map, Br1, is located just inside the opening to the interdomain cavity (*Figure 4B*). Br1 is modeled at nearly full (85%) occupancy and appears to make hydrogen-bonding contacts with the side chains of Arg105 from the N-terminal domain and Tyr473 from the C-terminal domain, both residues identified in our mutagenesis scan. Notably, the apo CylK model also contained a positive $F_o–F_c$ electron density peak at the Br1 site. Given the presence of chloride salts in the protein storage buffer, we modeled this site as a chloride ion in the apo model (*Figure 2B*). Br1 resides within 8 Å of the calcium-binding sites in the N-terminal domain, and the backbone carbonyl of Arg105 coordinates the central $Ca^{2+}$ ion (Ca2) in the β-roll motif of the N-terminal domain. This observation further supports an active role for both domains in the CylK-catalyzed reaction. Br1 is located within 5–6 Å of other side chains that form a portion of the solvent accessible channel nearest to the protein surface (*Figure 4B*), including a number of polar residues that could be implicated in activation of the resorcinol nucleophile and/or the alkyl chloride electrophile. Two residues, Asp440 and Leu438, undergo a rotamer change upon bromide binding. A second bromide peak, Br2, modeled at 48% occupancy, is located deep within the interdomain cavity. This ion makes contact with Ser318, Asn334, and Trp320, all contributed by the C-terminal domain. While residues surrounding both bromides are generally conserved in closely related CylK homologs, those associated with Br2 are not conserved in the C–O bond-forming family member BrtB (*Table 4*). This observation suggests that while Br2 residues might be relevant for paracyclophane-forming enzymes, they may not be necessary for all family members.

## Key catalytic residues are located at Br1 within the active site cleft

Having discovered that the CylK active site cleft has two potential alkyl halide-binding/-activating regions, we wondered if the two discrete native alkylation reactions catalyzed by CylK occurred at one or both bromide-binding sites. If only a single bromide site is catalytically active, we speculate that the monoalkylated intermediate **2** is likely released, rotates 180°, and must rebind the active site for the second alkylation event to occur (*Figure 4C*). To test this proposal, we individually mutagenized several residues closely associated with or appearing to directly bind either bromide, and examined their ability to perform the first and second alkylation reactions with native substrates **1** and **2**. Considering that the first alkylation is effectively faster than the second (*Nakamura et al., 2017*), likely due to substrate and/or product inhibition, we monitored the first alkylation by interrogating the 1 hr time point of the native reaction, when wild-type enzyme has fully transformed both equivalents of **1** to intermediate **2** (>99% conversion). Notably, substitution of the residues that comprise the Br1-binding site (Arg105, Tyr473) abolished CylK activity toward **1** at 1 hr, while mutations that disrupt Br2 (Ser318, Asn334) had no apparent effect (*Figure 4D*), retaining wild-type levels of activity. Variants targeting three additional residues located between the two bromide sites (Glu374, Leu438, Asp440) had very low or no activity and stood out as potential candidates for substrate binding or nucleophile activation due to their side chain functionalities and vicinity to Br1 on the cleft surface. Mutating Phe499 to Ala did not significantly reduce conversion.

We then examined the second alkylation reaction using a mixture of intermediate **2** containing a small amount of cylindrocyclophane F (~18%) as the substrate. Consistent with the results obtained for the first alkylation reaction, variants that disrupted Br1 did not have appreciable activity on intermediate **2**, while substitution of residues comprising Br2 maintained near wild-type activity (*Figure 4E*). Mutating Glu374, Leu438, and Asp440 also resulted in no significant activity toward **2** after 22 hr, while the Phe499 to Ala mutant retained appreciable activity as before. Of note, mutating Arg105 to Lys, and Asp440 to Asn resulted in no activity for either alkylation, suggesting that the size, hydrogen-bonding capacity, and/or $pK_a$ of these residues might be important for catalysis (*Figure 4—figure supplement 1*). However, mutating Tyr473 to Phe only moderately reduced activity and might suggest that the aromatic nature and/or size of this residue is critical for alkylation. Based on these results, we propose that Br1 is the site of alkyl chloride binding and both aromatic ring alkylation events. Additional work is required to determine if the residues that comprise Br2 assist in substrate binding,

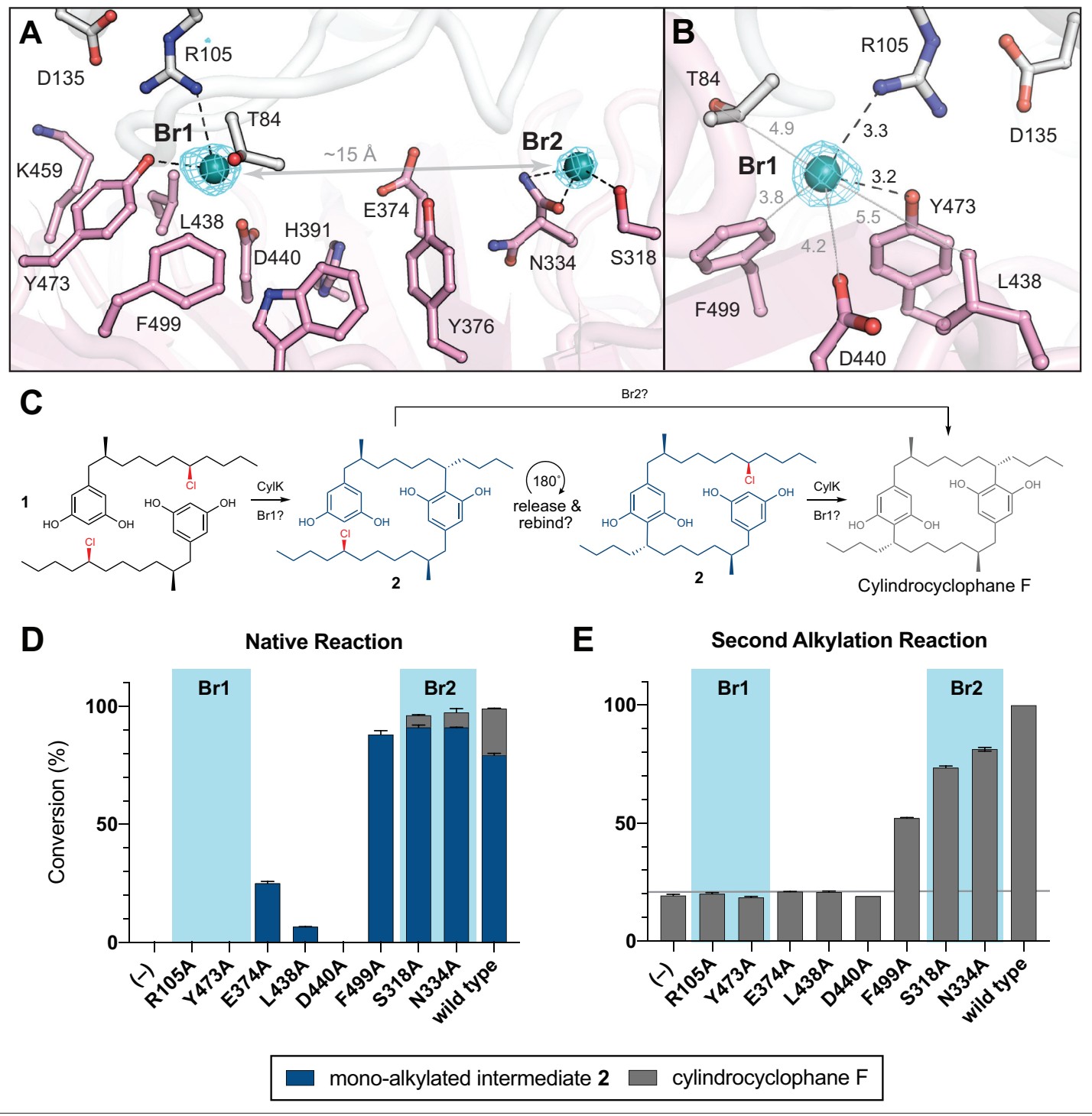

**Figure 4.** NaBr soak and mutagenesis of bromide-binding residues suggest a single site for catalysis. (**A**) A soak of NaBr into CylK crystals reveals two strong peaks in the anomalous difference electron density map (cyan mesh, contoured at 5.0 σ) within cavity 1. Bromide ion 1 (Br1) and 2 (Br2) are shown as teal spheres, and selected amino acids in the vicinity of each site are shown in stick format. The C-terminal domain is colored white, and the N-terminal domain is colored pink. (**B**) Alternate view of Br1 with anomalous difference electron density map, polar contacts shown as black dashed lines, and other distances shown as gray lines. All distances measured in angstroms. (**C**) Proposed alkylation scheme with potential roles for two bromide-binding sites in paracyclophane formation, or invoking release and rebinding of monoalkylated intermediate **2** for alkylation at a single bromide site. (**D**) End point activity at 1 hr of select mutants performing the native reaction with substrate **1**, highlighting the residues associated with Br1 or Br2. Product formation was quantified by high-performance liquid chromatography (HPLC), error bars represent the standard deviation from the mean of two biological replicates. (**E**) End point activity at 22 hr of select mutants performing the second alkylation reaction with intermediate **2**.

*Figure 4 continued on next page*

*Figure 4 continued*

The online version of this article includes the following source data and figure supplement(s) for figure 4:

**Source data 1.** Western blot of Strep-tagged CylK mutants (from left to right): R105A, R105K, S318A, N334A, E374A, L438A; and serial dilutions of purified wild-type CylK: 2.0, 0.5, and 0.1 µM.

**Source data 2.** Western blot of Strep-tagged CylK mutants (from left to right): D440A, D440N, Y473A, Y473F, F499A, wild-type; and serial dilutions of purified wild-type CylK: 2.0, 0.5, and 0.1 µM.

**Figure supplement 1.** Mutagenesis of active site cleft.

**Figure supplement 2.** Western blotting of Strep-tagged CylK mutants from soluble lysate with anti-Strep-horseradish peroxidase (HRP, IBA) in order to ensure mutant enzyme expression and solubility.

but it is clear from these data that they are not essential for catalysis. Furthermore, the site of catalysis at Br1 is located near the outermost portion of the solvent-exposed interdomain cavity; this is in agreement with previous work that demonstrated CylK's ability to accept non-native substrates with rigid substitutions on the resorcinol ring nucleophile (*Schultz et al., 2019*), which would necessitate a flexible or open active site.

## Molecular dynamics simulations suggest mode of substrate binding and activation

Having identified the site of alkylation and potential key residues, we sought to determine the specific mode of substrate binding and assess the relative contributions of specific side chains toward nucleophile (resorcinol) versus electrophile (alkyl chloride) activation. We began by computationally docking two equivalents of native substrate **1** into the active site of CylK (*Figure 5A*). The initial complexes were generated manually using the Br1 site to anchor the electrophilic alkyl chloride substituent of one equivalent of **1**. Previous DFT calculations performed on a model alkyl resorcinol substrate suggested that interaction between a resorcinol phenol substituent and a carboxylate hydrogen bond acceptor would enhance nucleophilicity of the aromatic ring (*Schultz et al., 2019*). Accordingly, potential interactions with key carboxylate residues Asp440 and Glu374 guided the placement of the nucleophilic resorcinol of the other equivalent of substrate **1**. Additionally, we oriented the nucleophilic and electrophilic carbon centers at an appropriate distance (<4 Å) and arrangement to accommodate the known stereochemical inversion of the substitution reaction. Two docked initial complexes were used, and both were run as restrained and unrestrained simulations. Restrained simulations included 1 kcal/mol·Å² restraints on all protein atoms to allow substrates to search the active site. All simulations consistently demonstrated that the active site cleft can accommodate two equivalents of substrate **1**, and furthermore, Asp440 and Glu374 maintained H-bonding contacts with the resorcinol nucleophile (*Figure 5B*). For the majority of the simulation time, the chloride-binding pocket formed by Arg105 and Tyr473 (Br1) maintained contact with the alkyl chloride electrophile (*Figure 5—figure supplement 1*, *Figure 5—figure supplement 2*). We conclude that Asp440 and Glu374 are likely resorcinol nucleophile-interacting residues, while Arg105 and Tyr473 bind and activate the alkyl chloride electrophile with hydrogen-bonding interactions. Furthermore, the source of CylK's exquisite regio- and stereoselectivity can be explained by the orientation in which the resorcinol nucleophile is held in close proximity to the backside of the carbon–chlorine (C–Cl) bond. Based on these results, we can propose a mechanism of alkylation that is distinct from chemical and enzymatic precedent (*Figure 5C*). Simulations of intermediate **2**, positioned for the second alkylation reaction, maintained similar alkyl chloride electrophile interactions; however, the resorcinol nucleophile did not remain at an appropriate distance or orientation for the reaction (*Figure 5—figure supplement 3*). More work is required to accurately model the second alkylation step; however, it is clear from our mutagenic work that the same residues are essential for both alkylation events.

## Bioinformatic analysis supports proposed residues implicated in alkyl chloride activation

We next sought to apply the functional insights gained from our structural analysis of CylK to improve our understanding of less closely related family members. Previously, bioinformatically identifying CylK-like enzymes was challenging because proteins encoding β-propeller motifs are common in publicly available protein databases, making it unclear which hits were true family members. With the

**Table 5.** Data collection and refinement statistics for Br-soaked *C. licheniforme* CylK structures.

| | Native | Br anomalous |
|---|---|---|
| **Data collection** | | |
| Space group | $C\,2\,2\,2_1$ | $C\,2\,2\,2_1$ |
| Wavelength (Å) | 0.97872 | 0.9184 |
| Cell dimensions | | |
| $a$, $b$, $c$ (Å) | 93.470, 138.656, 99.511 | 93.557, 138.726, 99.706 |
| α, β, γ (°) | 90.0, 90.0, 90.0 | 90.0, 90.0, 90.0 |
| Resolution (Å) | 50.0–1.52 (1.55–1.52) | 50–1.54 (1.57–1.54) |
| $R_{\text{merge}}$ | 0.076 (0.703) | 0.725 (4.755) |
| $R_{\text{pim}}$ | 0.029 (0.295) | 0.149 (1.140) |
| $I\,/\,\sigma I$ | 23.3 (2.14) | 24.0 (2.06) |
| $CC_{1/2}$ | 0.990 (0.843) | 0.978 (0.124) |
| Completeness (%) | 99.7 (98.9) | 100.0 (100.0) |
| Redundancy | 7.0 (5.5) | 24.2 (16.9) |
| | | |
| **Refinement** | | |
| Resolution (Å) | 46.78–1.52 | |
| No. reflections | 89,122 | |
| $R_{\text{work}}$ / $R_{\text{free}}$ | 0.1693/0.1978 | |
| No. atoms | 5437 | |
| Protein | 4989 | |
| Ligand/ion | 45 | |
| Water | 403 | |
| $B$-factors (Å$^2$) | | |
| Protein | 16.68 | |
| Ligand/ion | 23.03 | |
| Water | 23.77 | |
| RMS deviations | | |
| Bond lengths (Å) | 0.013 | |
| Bond angles (°) | 1.823 | |
| Molprobity clashscore | 1.64 (99th percentile) | |
| Rotamer outliers (%) | 0.76 | |
| Ramachandran favored (%) | 96.41 | |
| Ramachandran allowed (%) | 3.59 | |
| Ramachandran outlier (%) | 0.00 | |
| PDB accession code | 7ROO | |

*Values in parentheses are for highest-resolution shell.

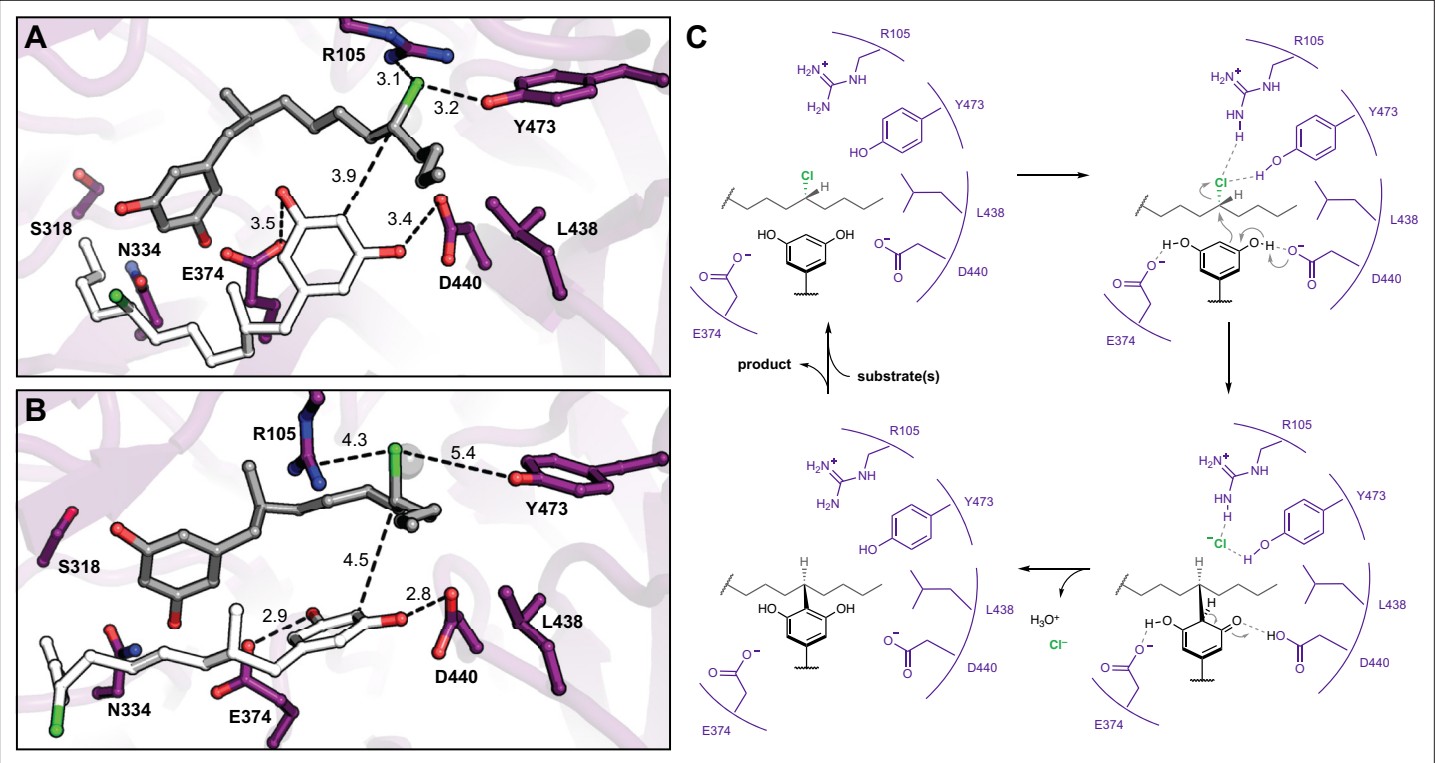

**Figure 5.** Molecular dynamics simulations reveal roles for key catalytic residues and enable a mechanistic proposal. (**A**) An energy-minimized docking model of CylK in complex with two chlorinated alkylresorcinol molecules in cavity 1 before and (**B**) after unrestrained molecular dynamics simulation. This analysis shows that both substrates can be accommodated while maintaining contact with one another and essential catalytic residues. Substrate molecules and selected amino acid side chains shown in stick format. (**C**) Proposed mechanism for a single cycle of the CylK-catalyzed Friedel–Crafts alkylation with key residues illustrated.

The online version of this article includes the following figure supplement(s) for figure 5:

**Figure supplement 1.** Molecular dynamics simulations of enzyme–substrate complexes manually docked into interdomain cavity support active site assignment in this region of CylK.

**Figure supplement 2.** Molecular dynamics simulations of intermediate product 2 complexes manually docked into interdomain cavity reveal a halide-binding pocket involving R105 and Y473.

**Figure supplement 3.** Molecular dynamics simulations of monoalkylated intermediate 2 positioned for the second alkylation reaction.

**Figure supplement 4.** Comparison of bound alkyl halides and their respective enzyme active sites.

knowledge that both protein domains of CylK are necessary for catalysis, we could now accurately locate CylK-like enzymes encoded in sequenced genomes. From a BLAST search, we identified over 700 proteins with >24% amino acid identity to CylK or BrtB (*Supplementary file 1*). That group was pared down to 286 unique enzyme sequences containing both N- and C-terminal domains found in CylK, and their relationship was assessed by constructing a maximum-likelihood phylogenetic tree (*Figure 6*).

We further analyzed these data by looking for the presence or absence of a partner CylC halogenase encoded in the same genome. A subset of CylK homologs that are mostly clustered together on the phylogenetic tree are found in organisms that also have a putative CylC halogenase. Of these CylK and CylC enzyme pairs, ~80% are co-localized in their respective genomes, hinting that they might work together within a biosynthetic pathway (*Supplementary file 2*). We found very few other types of halogenases co-localized with CylK homologs; those identified were of the iron(II) 2-(oxo)-glutarate (Fe$^{II}$/2OG)-dependent enzyme family but were also clustered with CylC halogenases. We examined each putative CylK sequence for the presence of the proposed alkyl chloride-activating residues Arg105 and Tyr473. Intriguingly, nearly all of the candidate CylK enzymes containing both key residues were from organisms that also encode a CylC halogenase (42 of 53 organisms), suggesting that they are likely using halogenated substrates and that Arg105 and Tyr473 are indeed important for

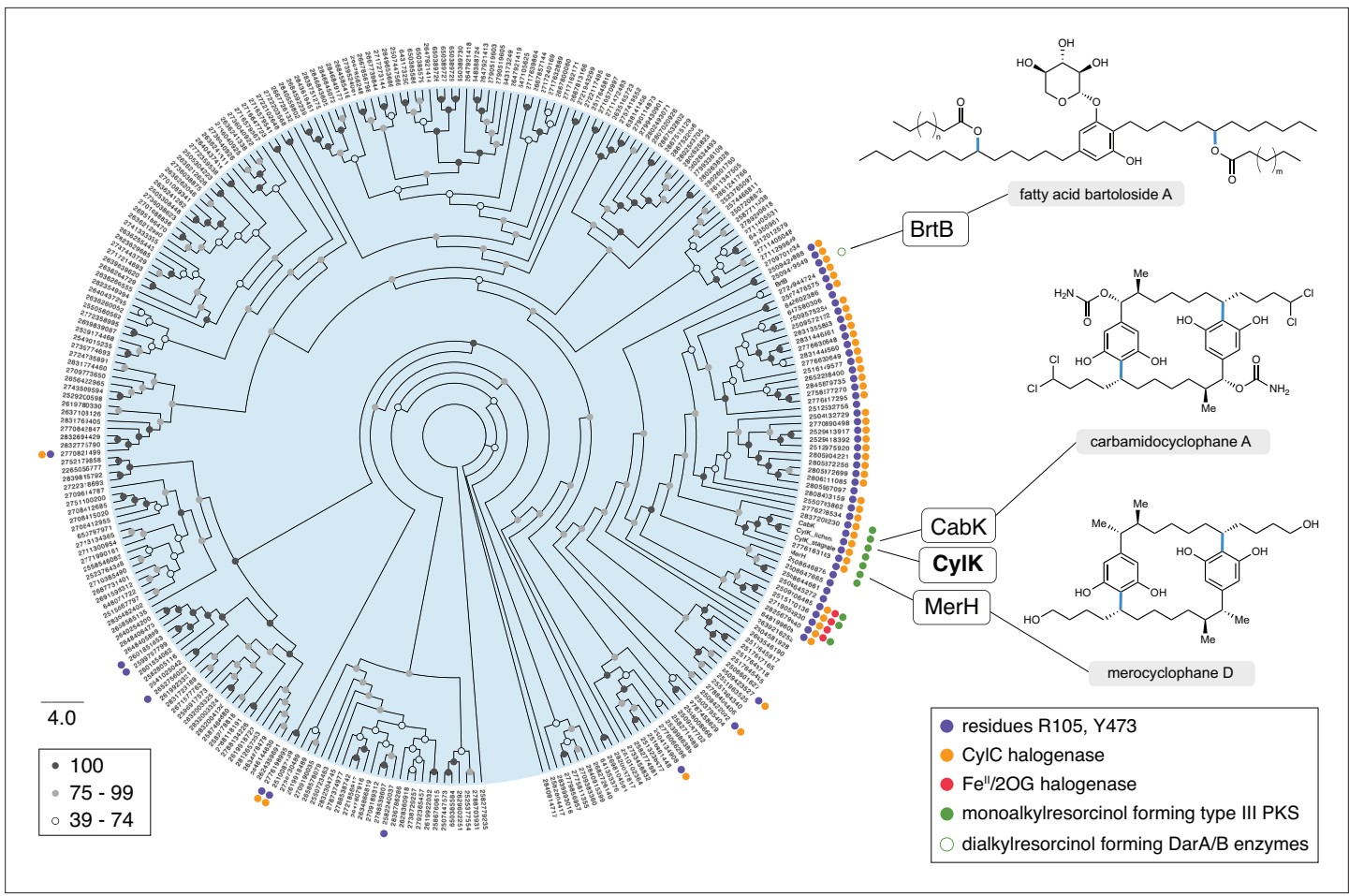

**Figure 6.** CylK homologs with partner CylC halogenases contain the proposed alkyl chloride-activating residues. Maximum-likelihood phylogenetic tree of protein sequences homologous to CylK and BrtB that contain both N- and C-terminal domains (>24% amino acid identity), highlighting the key conserved catalytic residues, clustered halogenases, and associated monoalkylresorcinol-forming enzymes. Dialkylresorcinol-forming enzymes (DarA/B-like) were not found except as expected with BrtB and the bartoloside-producing cluster, indicated as an unfilled green circle. Bootstrap values are shaded based on the legend. Natural products associated with select CylK homologs are displayed; the bonds highlighted in blue are constructed by their respective CylK.

The online version of this article includes the following figure supplement(s) for figure 6:

**Figure supplement 1.** Conservation mapping analysis of functionally annotated CylK homologs.

alkyl chloride activation. This subset of CylK homologs was aligned and visualized (*Figure 6—figure supplement 1*), which revealed that, while the resorcinol nucleophile-activating residue Asp440 was also highly conserved, Leu438, Glu374, and other active site residues are not highly prevalent within this subset. This demonstrates that the correlation between Arg105, Tyr473, and the presence of a CylC halogenase is not simply due to the overall similarity of this subset of enzymes.

Inspired by the biosynthetic logic of the *cyl* pathway, we also looked for co-localized monoalkylresorcinol (MAR)-forming enzymes encoded near putative CylK homologs in order to ascertain if resorcinols are likely to serve as their native nucleophilic substrates. To our knowledge, biosyntheses of MARs only involve type III PKS enzymes such as CylI (*Martins et al., 2019*). Surprisingly, we found only 10 CylI homologs clustered with CylK homologs, representing just 19% of the candidate enzymes that have Arg105 and Tyr473 residues and are expected to have alkyl chloride-activating activity. This observation suggests that a broader diversity of nucleophilic substrates might be used by these uncharacterized alkyl chloride-activating enzymes. Of note, dialkylresorcinol-forming enzymes such as those involved in bartoloside biosynthesis were not found in CylK-encoding gene clusters beyond the bartoloside gene cluster. The observation that the majority of CylK homologs do not have the proposed alkyl chloride-activating residues, partner halogenases, nor resorcinol forming enzymes,

indicates that this enzyme family may catalyze other reactions not involving alkyl halides or resorcinols. This information will enable future studies by prioritizing organisms that might produce more distantly related natural products and/or CylK-like enzymes with alternate substrate scopes.

## Discussion

Our results provide a structural basis for understanding the unusual enzymatic Friedel–Crafts alkylation catalyzed by CylK in cylindrocyclophane biosynthesis. This structural model, together with supporting biochemical experiments and bioinformatic analyses, has provided fundamental insight into this intriguing reaction that represents the only known example of aromatic ring alkylation with an alkyl halide electrophile. The molecular knowledge we have gained may be applied to access and engineer novel biocatalysts, design chemocatalysts, and discover new natural products that may be candidate therapeutics and/or play important physiological roles in cyanobacteria. We note that a recently published paper also described the structure of CylK and reached similar conclusions (*Wang et al., 2022*).

We identified the active site of CylK at the interface of its N- and C-terminal domains. Together with previously characterized enzymes, our results show how structurally related β-propeller enzymes have divergently adapted this fold for catalytic function. In two metalloenzymes, carotenoid cleavage dioxygenase (CCD) (*Messing et al., 2010*) and nitrous oxide reductase (NOR) (*Pomowski et al., 2011*), an active site metallocofactor is lodged on the top face of the propeller and coordinated by His side chains from inner β-strands of the fold (*Figure 2—figure supplement 4*). Like CylK, both of these enzymes have a capping motif on the top face of the propeller. CCD contains more extensive internal loops than CylK, and these adopt structured helix-loop motifs that bury the top face of the propeller. NOR is a di-domain dimeric enzyme that employs an electron transfer domain from an adjacent monomer to shield the catalytic metallocofactor in the β-propeller motif. In both, these domain arrangements create a favorable environment for substrate binding and catalysis, as we observe in CylK (*Figure 2—figure supplement 6*).

Both domains of CylK have several bound calcium ions, a row of which is near the proposed alkyl chloride-binding site. However, the closest $Ca^{2+}$ ion is approximately 8 Å from this pocket. Although we have previously demonstrated a functional requirement for calcium (*Nakamura et al., 2017*), our structural work suggests that a direct role in catalysis is unlikely. Removing calcium might cause a significant conformational change that precludes access to or alters the architecture of the active site as the treatment of CylK with ethylenediaminetetraacetic acid (EDTA) was observed to cause a shift in oligomeric state from monomer to a dimer. This might be indicative of larger structural changes within each unit of CylK that are incompatible with catalysis. It is unknown if calcium plays a regulatory role in vivo, although it is a component of the cyanobacterial BG-11 growth medium.

We propose a plausible mechanism for CylK catalysis that invokes positioning the reactive partners in close proximity for a concerted $S_N2$-like reaction, consistent with the known inversion of configuration at the alkyl chloride stereocenter upon substitution (*Figure 5C*). In particular, we suggest that the side chain carboxylates of Asp440 and Glu374 hydrogen bond with the two phenol substituents of the resorcinol nucleophile. By deprotonating one phenol, the essential residue Asp440 could enhance the nucleophilicity of the reactive carbon. Analogous deprotonation of a resorcinol has been invoked as a critical step in an enzymatic Friedel–Crafts acylation reaction with acetyl arene electrophiles, lending support to our proposal (*Pavkov-Keller et al., 2018*). Simultaneously, we hypothesize that Arg105, Tyr473, Phe499, and the hydrophobic portion of Thr84 form an alkyl chloride-binding pocket as observed in the bromide-soaked CylK structure. Hydrogen-bonding interactions between the polar residues and the alkyl chloride could weaken the C–Cl bond, reducing the overall activation energy barrier for substitution. Interestingly, mutating Tyr473 to Phe only slightly reduces alkylation activity, which suggests that the alkyl chloride Tyr473 interaction might resemble an anion–pi interaction (*Schottel et al., 2008*; *Garau et al., 2003*). While there is precedent for alkyl halide–arene interactions (*Parisini et al., 2011*), they are less thoroughly studied than anion–pi interactions. Alternatively, if Tyr473 provides a key hydrogen bond, the Phe mutant may partially compensate by forming an alkyl halide–pi interaction. In either case, this alkyl chloride-binding pocket is reminiscent of the active sites of the *S*-adenosylmethionine (SAM)-dependent chlorinase (*Eustáquio et al., 2008*) and fluorinase (*Dong et al., 2004*) enzymes. In these structures, similar ensembles of polar and aliphatic residues bind and stabilize a halide anion as it reacts with SAM in a $S_N2$ substitution reaction that

resembles the reverse of the transformation catalyzed by CylK (**Figure 5—figure supplement 4A and B**). Interestingly, our proposed CylK alkyl chloride-binding mode is distinct from that of the well-studied haloalkane dehalogenase enzyme, which primarily uses the indole N–H bonds of adjacent Trp residues to activate a primary alkyl chloride for substitution by the side chain carboxylate of an active site Asp via hydrogen bonding (**Figure 5—figure supplement 4C**; **Verschueren et al., 1993**). We speculate that the differences between these two enzymes arise from their distinct evolutionary histories as well as the decreased hydrophobicity and increased reactivity of a primary alkyl chloride substrate. Following alkylation, the chloride anion product diffuses out of the active site, and the aromaticity of the resorcinol is restored upon proton transfer to water, perhaps via His391. Finally, intermediate **2** is released from the active site, must reorient 180°, and rebind for the second intramolecular alkylation to occur in the same fashion. This reorientation likely requires significant energy to disrupt protein–substrate interactions and might suggest active site conformational changes occur to accommodate intermediate **2** in the catalytically competent orientation; however, this warrants further investigation. Although we cannot rule out a radical mechanism, the architecture of the active site and lack of radical initiation sources suggests that one-electron processes are unlikely.

We have emphasized the distinction between the proposed resorcinol nucleophile-activating residues and those that likely activate the alkyl chloride electrophile in order to compare CylK with the divergent C–O bond-forming family member, BrtB (30% amino acid identity, 46% similarity). We hypothesize that both enzymes utilize a similar strategy to enhance the reactivity of their alkyl chloride substrates, although it remains to be determined if BrtB is similarly stereoinvertive. In agreement with this proposal, BrtB contains Arg105 and Tyr473 equivalents that may play a role in alkyl chloride activation. In place of the nucleophile-activating Asp440, BrtB contains a functionally similar Glu, and notably, Glu374 is substituted with a positively charged Arg. These two residues might interact with carboxylate nucleophiles, although the precise roles of these residues in BrtB's distinct reactivity remain to be determined. Interestingly, a carboxylate nucleophile would be predominantly deprotonated at physiological pH, suggesting that nucleophile activation might be unnecessary in this enzyme. Regardless, the divergent reactivity of CylK and BrtB hints that other relatives might perform unique chemistry and may even be amenable to protein engineering efforts to alter their substrate scopes and reactivity.

The reaction catalyzed by CylK represents a unique approach for enzymatic C–C bond formation. Other biological strategies to form analogous arene–alkyl C–C bonds involve radical intermediates that must be precisely held and oriented in their enzyme active sites in order to prevent side reactions. Examples include the radical SAM and Fe$^{II}$/2OG enzymes involved in streptide (**Schramma et al., 2015**) and etoposide (**Lau and Sattely, 2015**) biosynthesis, respectively. In contrast to CylK, these enzymes form bonds between arenes and unactivated carbon centers. Because the target sp$^3$ coupling sites of their substrates are inherently unactivated, these alkylating enzymes must stringently discriminate between the alkyl C–H coupling site and its chemically equivalent neighbors. In contrast, the CylK-catalyzed reaction leverages the preinstalled alkyl chloride substituent in order to direct reactivity. Therefore, the cylindrocyclophane biosynthetic logic can be described as dividing the task of determining reaction specificity between CylK and its partner halogenase, CylC. This novel halogenase family likely has strict requirements for substrate positioning (**Matthews et al., 2009**), although it remains to be seen if CylC is promiscuous. This feature of distributing selectivity and alkylation across two enzymes might rationalize why wild-type CylK can accept multiple non-native substrates, whereas active site specificity for unactivated coupling partners might be more stringent.

CylK performs a challenging chemical reaction without precedent in biology. By elucidating its structure, we have begun to unravel the molecular details underlying the use of alkyl halides as cryptic intermediates in natural product biosynthesis. This enzymatic Friedel–Crafts alkylation might also find application in constructing diverse alkyl–arene C–C bonds outside of natural pathways. With structural information about CylK, notably the identification of active site residues, efforts to further expand its substrate specificity and improve its stability via enzyme engineering can begin in earnest. In addition to its unique reactivity, the large size and solvent accessibility of CylK's active site suggest that it might be engineered to accept diverse and bulky substrates. Furthermore, our structure and proposed mechanism have increased our fundamental understanding of enzymatic interactions with alkyl halides and may inspire the design of biomimetic chemocatalysts that engage secondary alkyl halides in substitution reactions (**Park et al., 2017**; **Brak and Jacobsen, 2013**). Finally, armed with the

ability to functionally annotate CylK-like genes, we can prioritize orphan biosynthetic gene clusters and producing organisms in order to discover additional structurally unique natural products.

# Materials and methods

## Key resources table

| Reagent type (species) or resource | Designation | Source or reference | Identifiers | Additional information |
|---|---|---|---|---|
| Gene (*Cylindrospermum licheniforme* ATCC 29412) | CylK | GenBank | ARU81125.1 | |
| Strain, strain background (*Escherichia coli*) | BL21(DE3) | Invitrogen | C6000-03 | |
| Strain, strain background (*E. coli*) | BL21 Gold CodonPlus (DE3) RIL | Agilent | 230245 | |
| Peptide, recombinant protein | Strep-Tactin HRP, anti-Strep-HRP (*Streptomyces avidinii*) | IBA Life Sciences | 2-1502-001 | (1:25,000) |
| Recombinant DNA reagent | pET-His6-Sumo-CylK-Strep (plasmid) | *Schultz et al., 2019* | | |
| Recombinant DNA reagent | pPR-IBA1-CylK (plasmid) | *Nakamura et al., 2017* | | |
| Recombinant DNA reagent | pPR-IBA1-CylK K78E (plasmid) | This work, constructed by GENEWIZ | | Replaced AAG with GAG at position 232 |
| Recombinant DNA reagent | pPR-IBA1-CylK D96A (plasmid) | This work, constructed by GENEWIZ | | Replaced GAT with GCT at position 286 |
| Recombinant DNA reagent | pPR-IBA1-CylK R105A (plasmid) | This work, constructed by GENEWIZ | | Replaced CGT with GCC at position 313 |
| Recombinant DNA reagent | pPR-IBA1-CylK Y473A (plasmid) | This work, constructed by GENEWIZ | | Replaced TAT with GCT at position 1,417 |
| Recombinant DNA reagent | pPR-IBA1-CylK S265A (plasmid) | This work, constructed by GENEWIZ | | Replaced AGT with GCC at position 793 |
| Recombinant DNA reagent | pPR-IBA1-CylK S266A (plasmid) | This work, constructed by GENEWIZ | | Replaced TCC with GCC at position 796 |
| Recombinant DNA reagent | pPR-IBA1-CylK T267A (plasmid) | This work, constructed by GENEWIZ | | Replaced ACC with GCT at position 799 |
| Recombinant DNA reagent | pPR-IBA1-CylK S324A (plasmid) | This work, constructed by GENEWIZ | | Replaced AGT with GCT at position 970 |
| Recombinant DNA reagent | pPR-IBA1-CylK K377A (plasmid) | This work, constructed by GENEWIZ | | Replaced AAA with GCC at position 1129 |
| Recombinant DNA reagent | pPR-IBA1-CylK T379A (plasmid) | This work, constructed by GENEWIZ | | Replaced ACC with GCT at position 1135 |
| Recombinant DNA reagent | pPR-IBA1-CylK D502A (plasmid) | This work, constructed by GENEWIZ | | Replaced GAC with GCC at position 1504 |
| Recombinant DNA reagent | pPR-IBA1-CylK E504A (plasmid) | This work, constructed by GENEWIZ | | Replaced GAA with GCC at position 1510 |
| Recombinant DNA reagent | pPR-IBA1-CylK L558A (plasmid) | This work, constructed by GENEWIZ | | Replaced TTA with GCT at position 1672 |
| Recombinant DNA reagent | pPR-IBA1-CylK D559A (plasmid) | This work, constructed by GENEWIZ | | Replaced GAT with GCT at position 1675 |
| Recombinant DNA reagent | pPR-IBA1-CylK D562A (plasmid) | This work, constructed by GENEWIZ | | Replaced GAT with GCC at position 1684 |
| Recombinant DNA reagent | pPR-IBA1-CylK T616A (plasmid) | This work, constructed by GENEWIZ | | Replaced ACG with GCT at position 1846 |
| Recombinant DNA reagent | pPR-IBA1-CylK T618A (plasmid) | This work, constructed by GENEWIZ | | Replaced ACT with GCT at position 1852 |

*Continued on next page*

*Continued*

| Reagent type (species) or resource | Designation | Source or reference | Identifiers | Additional information |
|---|---|---|---|---|
| Recombinant DNA reagent | pPR-IBA1-CylK D440A (plasmid) | This work, constructed by GENEWIZ | | Replaced GAC with GCC at position 1318 |
| Recombinant DNA reagent | pPR-IBA1-CylK L438A (plasmid) | This work, constructed by GENEWIZ | | Replaced CTT with GCT at position 1312 |
| Recombinant DNA reagent | pPR-IBA1-CylK E374A (plasmid) | This work, constructed by GENEWIZ | | Replaced GAA with GCT at position 1120 |
| Recombinant DNA reagent | pPR-IBA1-CylK F499A (plasmid) | This work, constructed by GENEWIZ | | Replaced TTT with GCC at position 1495 |
| Recombinant DNA reagent | pPR-IBA1-CylK N334A (plasmid) | This work, constructed by GENEWIZ | | Replaced AAC with GCC at position 1000 |
| Recombinant DNA reagent | pPR-IBA1-CylK S318A (plasmid) | This work, constructed by GENEWIZ | | Replaced AGC with GCC at position 952 |
| Recombinant DNA reagent | pPR-IBA1-CylK R105K (plasmid) | This work, constructed by GENEWIZ | | Replaced CGT with AAG at position 313 |
| Recombinant DNA reagent | pPR-IBA1-CylK Y473F (plasmid) | This work, constructed by GENEWIZ | | Replaced TAT with TTT at position 1417 |
| Recombinant DNA reagent | pPR-IBA1-CylK D440N (plasmid) | This work, constructed by GENEWIZ | | Replaced GAC with AAC at position 1318 |
| Software, algorithm | HKL2000 | *Otwinowski and Minor, 1997* | http://www.hkl-xray.com/hkl-2000; RRID:SCR_015547 | |
| Software, algorithm | Phaser | *McCoy et al., 2007* | https://www.phenix-online.org/documentation/reference/phaser.html; RRID:SCR_014219 | |
| Software, algorithm | Phenix | *Liebschner et al., 2019* | https://www.phenix-online.org/; RRID:SCR_014224 | |
| Software, algorithm | Coot | *Emsley and Cowtan, 2004* | http://www2.mrc-lmb.cam.ac.uk/personal/pemsley/coot/; RRID:SCR_014222 | |
| Software, algorithm | CCP4 (FFT, CAD) | *Winn et al., 2011* | http://www.ccp4.ac.uk/; RRID:SCR_007255 | |
| Software, algorithm | Refmac5 | *Vagin et al., 2004* | http://www.ccp4.ac.uk/html/refmac5/description.html; RRID:SCR_014225 | |
| Software, algorithm | PyMOL | The PyMOL Molecular Graphics System, version 2.0 Schrödinger, LLC | http://www.pymol.org/; RRID:SCR_000305 | |

## Protein expression and purification

CylK was expressed and purified for crystallographic characterization according to a modified literature method (*Schultz et al., 2019*). Following overexpression in *Escherichia coli* BL21(DE3) as a SUMO fusion construct, cell pellets were resuspended in lysis buffer (20 mM Tris pH 8.0, 500 mM NaCl, 10 mM MgCl$_2$, 10 mM CaCl$_2$) supplemented with an EDTA-free Pierce Protease Inhibitor Tablet. Cells were lysed using a continuous-flow homogenizer (Avestin EmulsiFlex-C3). The pellet was collected by centrifugation (20,000 × *g* for 30 min) and resuspended in lysis buffer supplemented with 6 M urea and 5 mM imidazole. The denatured enzyme solution was clarified twice by centrifugation (20,000 × *g* for 15 min), and the supernatant was applied to 20 mL of Ni NTA resin. The column was washed with lysis buffer containing 6 M urea and 5 mM imidazole. Bound species were eluted with lysis buffer containing 6 M urea and 250 mM imidazole. All subsequent steps were carried out at 4°C. The enzyme was refolded by sequentially dialyzing this solution against buffer 1 (lysis buffer supplemented with 3 M urea) overnight, buffer 2 (lysis buffer lacking urea) for 4 hr, and buffer 3 (lysis buffer supplemented with 5% glycerol) for 4 hr. Following dialysis, the refolded enzyme solution was centrifuged (100,000 × *g* for 30 min), and the supernatant was collected and incubated overnight with ULP

Protease (1:25 w/w). Then, following a subtractive Ni NTA step, the enzyme was applied to a Superdex 200 30/100 gel filtration column (GE Healthcare) equilibrated in protein storage buffer (20 mM HEPES pH 7.8, 50 mM NaCl, 10 mM $MgCl_2$, 10 mM $CaCl_2$, 10% glycerol). The fractions containing pure CylK were pooled, concentrated (500–700 µM, determined by extinction coefficient and absorbance at 280 nm), and flash frozen in liquid nitrogen.

## Crystallization and structure determination

Prior to crystallization trials, purified CylK protein in storage buffer was combined with 5-undecylbenzene-1,3-diol **3**, a substrate analog synthesized as described previously (*Schultz et al., 2019*) and dissolved in protein storage buffer containing 22% DMSO. The final protein solution contained 10 mg/mL CylK and 1.66 mM of **3**. Crystals were obtained by using the hanging drop vapor diffusion method in 2 µL drops mixed in 1:1 ratio with a precipitant solution of 1.8 M sodium malonate, pH 7.0. To aid in obtaining phase information, the native Ca(II) sites in CylK crystals were substituted with a lanthanide via soak in a solution of 2.0 M sodium malonate, 100 mM terbium(III) chloride for 8 min at room temperature. The Tb-soaked crystals were transferred to a cryoprotectant solution containing 2.0 M sodium malonate supplemented with 20% ethylene glycol, mounted on rayon loops, and flash frozen by direct plunge into liquid nitrogen. Diffraction datasets were collected at 1.6314 Å (Tb anomalous) and 0. 97872 Å (native) on these crystals at beamlines 23-ID-B (General Medical Sciences and Cancer Institutes Collaborative Access Team [GM/CA-CAT], Advanced Photon Source [APS]) and 21-ID-F (Life Sciences Collaborative Access Team [LS-CAT], APS), respectively. Diffraction datasets were processed with the HKL2000 software package (*Otwinowski and Minor, 1997*). The structure was solved by using the MR-SAD method. PHASER-EP (*McCoy et al., 2007*; *Read and McCoy, 2011*), as implemented within the PHENIX software package (*Liebschner et al., 2019*), was used to calculate initial phases. A polyalanine model generated from the core β-propeller domain of *Staphylococcus cohnii* streptogramin B lyase (PDB accession code 2QC5, with all loops and secondary structure connections truncated) (*Lipka et al., 2008*) was used as the initial search model for MR. PHENIX.autobuild generated an initial model containing 504 residues in 15 chain fragments with initial $R_{work}/R_{free}$ values of 29.9%/37.4%. The model was iteratively improved via refinement in Refmac5 against a native dataset and model building in Coot (*Emsley and Cowtan, 2004*), yielding $R_{work}/R_{free}$ values of 16.4%/19.8% (*Table 1*). The final model contains residues 7–45, 49–392, and 413–662 with 11 $Ca^{2+}$ ions, 2 $Mg^{2+}$ ions, 1 $Cl^-$ ion, and 339 water molecules. Tb ions were not modeled because they were not present at high occupancy. Ramachandran analysis (*Liebschner et al., 2019*) indicates a single outlier, Val620. Although **3** was required for CylK crystallization, and weak electron density resembling the compound could be found at several points near the surface of the protein, likely mediating crystal lattice contacts, we could not confidently model the alkyl resorcinol. No density for **3** could be identified in cavity 1, the putative active site of CylK.

To identify possible alkyl chloride-binding sites within the protein, CylK crystals were soaked in 2 M sodium malonate with 500 mM NaBr for up to 1 min. The Br-soaked crystals were transferred to a cryoprotectant solution consisting of the soak solution supplemented with 20% ethylene glycol. Diffraction datasets were collected at 0.9184 Å (Br anomalous) and 0.97872 Å (native) on these crystals at beamlines 23-ID-B (GM/CA-CAT, APS) and 21-ID-F (LS-CAT, APS), respectively. Phase information was obtained by MR using the CylK model as the initial search model in Phaser MR, implemented within CCP4. The model was iteratively built and refined in Coot and Refmac5, respectively, resulting in a $R_{work}/R_{free}$ of 16.9%/19.8% (*Table 5*). The final model contains residues 7–393 and 410–667 with 10 $Ca^{2+}$ ions, 2 $Mg^{2+}$ ions, 1 $Na^+$ ion, 4 $Br^-$ ions, and 403 water molecules. Anomalous maps were generated using CAD and FFT programs implemented within CCP4. Figures were generated in PyMOL (Schrödinger, LLC). The information in *Tables 1 and 5* was generated using HKL2000 (data collection statistics), Refmac5 (refinement statistics including resolution limits, $R_{work}/R_{free}$, rms deviations), the PBD validation server (no. of atoms), and PHENIX (*B*-factors, Molprobity clashscore, rotamer outliers, Ramachandran statistics) software packages.

## Mutant enzyme activity screening

CylK mutants were expressed and assayed for activity according to a modified literature method (*Schultz et al., 2019*). Point mutants in plasmid pPR-IBA1-CylK were synthesized and sequence verified by GENEWIZ, South Plainfield, NJ. Enzyme activity assays using substrate pair **3 + 4** were carried

out as follows. Chloro-SNAc electrophile **4** was accessed as described previously (*Schultz et al., 2019*). Respective mutant, wild-type, and negative control plasmids (Addgene #31122) were freshly transformed into electrocompetent *E. coli* BL21 Gold CodonPlus (DE3) RIL cells (Agilent), and an individual colony was used to directly inoculate 1 mL cultures of LB medium supplemented with 100 μg/mL ampicillin or carbenicillin and 34 μg/mL chloramphenicol in a 96 deep well plate (VWR); sterility was maintained with standard techniques and a gas-permeable rayon film (VWR). The liquid culture plate was incubated for 5 hr at 37°C and 250 rpm shaking. After 5 hr, and confirming visible growth in each well, the liquid culture plate was cooled to 15°C, maintaining shaking. Following an additional 30–45 min of incubation, protein expression was induced with 250 μM IPTG. The cultures were incubated for 4 hr at 15°C with 250 rpm shaking. Following expression, 10 μL of each culture was removed from each well and subcultured for liquid culture RCA-based DNA sequencing to confirm identity and rule out culture cross-contamination. The remaining cultures were concentrated ~10× by centrifugation (3220 × *g* for 10 min) and resuspended in ~200 μL of spent media supernatant. Reactions were initiated in opaque 96-well plates (Costar) by combining 100 μL of concentrated cell suspension with 2 μL of lysozyme mix (9 mg lysozyme and 1 mg EDTA-free Pierce Protease Inhibitor Tablet per 500 μL of 500 mM Tris, 50 mM EDTA, pH 8.0) and monoalkylation substrates in DMSO for a final concentration of nucleophile **3** at 150 μM, electrophile **4** at 300 μM, and DMSO at 3.4%. Reaction plates were sealed with aluminum seals (VWR) and incubated at 37°C with 190 rpm shaking for 20 hr. Reactions were quenched with the addition of 1:1 methanol/acetonitrile (200 μL) and centrifuged (3220 × *g* for 15 min). The supernatant was then subjected to liquid chromatography-mass spectrometry (LC-MS) to measure ion abundance of product **5** ($C_{31}H_{53}ClNO_4S$ [M + H$^+$] = 536.3768 ± 5 ppm) and starting material **4** ($C_{14}H_{26}ClNO_2S$ [M + H$^+$] = 308.1446 ± 5 ppm). LC-MS conditions were as published previously (*Schultz et al., 2019*). Reactions were repeated in biological triplicate.

Enzyme activity on native substrates **1** or **2** was assayed as follows. Substrate **1** and intermediate **2** were accessed chemoenzymatically as described previously (*Nakamura et al., 2017*). Care was taken to minimize glycerol in substrate stocks because it was determined to inhibit CylK activity. Respective mutant and wild-type plasmids were freshly transformed into electrocompetent *E. coli* BL21 Gold CodonPlus (DE3) RIL cells (Agilent), and 5–10 colonies were used to directly inoculate 100 mL cultures in LB medium supplemented with 100 μg/mL ampicillin or carbenicillin and 34 μg/mL chloramphenicol. The liquid cultures were incubated at 37°C and 190 rpm shaking. At OD400 = 0.4, the cultures were cooled to 15°C, maintaining shaking. Following an additional 1 hr, protein expression was induced with 250 μM IPTG. The cultures were then incubated for 4 hr at 15°C with 190 rpm shaking. Following expression, cell pellets were collected by centrifugation (3220 × *g* for 10 min) and resuspended in ~2 mL of assay buffer (20 mM HEPES pH 7.8, 100 mM NaCl, 10 mM MgCl$_2$, 5 mM CaCl$_2$). Cell concentration was normalized across mutants by measuring OD600 and diluting with an appropriate amount of assay buffer. A 600 μL aliquot of each mutant was lysed by sonication (Branson, 25% amplitude, 1 min) on ice. The soluble fraction was collected by centrifugation (16,100 × *g* for 20 min) and analyzed by anti-Strep-HRP (IBA) Western blotting to confirm mutant solubility and approximate protein concentration (*Figure 4—figure supplement 2*). Reactions were initiated by combining 4.3 μL of **1** or **2** with 25.7 μL of soluble lysate for a final concentration of **1** at 400 μM and 4% DMSO, and **2** at 200 μM and 2% DMSO. Reactions were sealed and incubated at 37°C for 1 or 20 hr. Reactions were quenched with the addition of 1:1 methanol/acetonitrile (60 μL) and centrifuged (6000 × *g* for 15 min). The supernatant was then subjected to analytical high-performance liquid chromatography (HPLC) to determine percent conversion as described previously. Reactions were repeated in biological duplicate, and then repeated again on a second day to minimize any potential variation. Results were consistent on both days.

## Molecular dynamics simulations

Coordinates for the native substrates **1** and **2** were prepared in ACEDRG (*Long et al., 2017*) from MOL file descriptions of the molecules generated in ChemDraw. Two substrate molecules were docked manually into cavity 1 of CylK using Br1 to place the alkyl chloride component and interactions with essential amino acids to guide placement of the second substrate. The resulting CylK complexes were prepared for molecular dynamics (MD) simulations using the Xleap module of AmberTools (*Case, 2020*). Substrates were parameterized using Antechamber (*Wang et al., 2001*; *Wang et al., 2004*) in AmberTools, while the FF14SB forcefield (*Maier et al., 2015*) was used for protein atoms. Complexes

were solvated in a 10 Å octahedral box of TIP3P water (*Jorgensen et al., 1983*). Sodium and chloride ions were added to neutralize the complexes and achieve a final concentration of 150 mM NaCl. All minimizations and production runs were performed with Amber20 (*Case, 2020*). Minimization was performed using 5000 steps of steepest descent and 5000 steps of conjugate gradient. In the first round, 500 kcal/mol·Å$^2$ restraints on all protein, ligand atoms, as well as the metal ion cofactors. Restraints were reduced to 100 kcal/mol·Å$^2$ for an additional round of solvent minimization. Next, restraints retained only ligands and metal ions for a third round of minimization. Finally, all restraints were removed for the last stage of minimization. After minimization, a 100 ps run was used to heat complexes from 0 to 300 K using constant volume periodic boundaries with 10 kcal/mol·Å$^2$ restraints on all solute atoms. Equilibration was performed using two 10 ns runs (10 kcal/mol·Å$^2$ and 1 kcal/mol·Å$^2$ restraints) on protein, ligand, and metal ions. Following equilibration, simulations were run as either restrained or unrestrained. For restrained simulations, we obtained 100 ns trajectories with 1 kcal/mol·Å$^2$ restraints on protein atoms. Unrestrained simulations (100 ns) were run with all restraints removed.

## Bioinformatic analysis

The CylK (GenBank: ARU81125.1) and BrtB (GenBank: AOH72618.1) protein sequences were independently used in a Basic Local Alignment Search Tool (BLAST) search of the Joint Genome Institute-Integrated Microbial Genomes & Microbiomes (IMG-JGI; *Chen et al., 2021*) database of all isolates, metagenome-assembled genomes (MAGs), and single-amplified genomes (SAGs). Each BLAST search resulted in 500 hits (>24% amino acid sequence identity) and were merged to form a nonredundant list of 715 enzyme candidates (*Supplementary file 1*). Parent enzyme sequences (CylK, BrtB) and likely CylK homologs (MerH, CabK, CylK from *Cylindrospermum stagnale*) were added to this dataset. A targeted BLAST search of the genomes from this list was performed to identify all putative halogenases (CylC-like, Fe$^{II}$/2OG, SAM-dependent, flavin-dependent, and haloperoxidase) and resorcinol-forming enzymes (CylI, BrtC, BrtD). All non-CylC halogenases were confirmed to contain key catalytic residues and examined for co-localization with CylK. All candidate CylC halogenases were included, although the overwhelming majority (63 of 78 enzymes) were fewer than 18 genes away from CylK (*Supplementary file 2*). CylC's catalytic residues remain unknown. MAR-forming CylI homologs were included if clustered with CylK; 11 of 54 candidate enzymes were less than 12 genes away from CylK (*Supplementary file 2*). Next, candidate CylK enzymes were filtered by constructing an individual MUSCLE alignment (*Edgar, 2004*) of each hit to the CylK parent sequence and identifying at least 135 amino acid residues before the C-terminal domain. Lys240 or its aligned equivalent was defined as the junction point between protein domains. This analysis resulted in 286 unique protein sequences (*Supplementary file 2*) that were aligned (MUSCLE, EMBL-EBI), sites containing gaps at ≥90% of sequences were removed, and a maximum-likelihood phylogenetic tree was constructed using IQ-TREE v1.6.12 (*Nguyen et al., 2015*) and visualized with FigTree v1.4.4. The best-fit model (VT + F + G4) was selected by ModelFinder (*Kalyaanamoorthy et al., 2017*). Bootstrap values were calculated using the UFBoot2 method (*Hoang et al., 2018*). These candidate enzymes were analyzed for the presence or absence of Arg105 and Tyr473 residues by constructing an individual MUSCLE alignment of each hit to the CylK parent sequence. The results were manually mapped onto the phylogenetic tree. The subset CylK homologs with both residues were MUSCLE aligned and visualized with ConSurf (*Ashkenazy et al., 2016*). The code used to perform iterative MUSCLE alignments is available at https://github.com/nbraffman/CylK-homologs, (copy archived at swh:1:rev:732a38e-f8ab73988203ea1933158238ea90361b0; *Braffman, 2021*).

## Acknowledgements

We gratefully acknowledge use of the resources of the Advanced Photon Source, a US Department of Energy (DOE) Office of Science User Facility operated for the DOE Office of Science by the Argonne National Laboratory under Contract DE-AC02-06CH11357. Use of Life Sciences Collaborative Access Team Sector 21 was supported by the Michigan Economic Development Corporation and the Michigan Technology Tri-Corridor (Grant 085P1000817). The National Institute of General Medical Sciences and National Cancer Institute Structural Biology Facility at the Advanced Photon Source (GM/CA@APS) has been funded in whole or in part with federal funds from the National Cancer Institute (ACB-12002) and the National Institute of General Medical Sciences (AGM-12006,

P30GM138396). The Eiger 16M detector at GM/CA@APS was funded by NIH Grant S10 OD012289. We thank Andrew J Mitchell and Jonathan A Bergman for their excellent technical assistance. This work was supported by NSF Grants 1454007 and 2003436, the Cottrell Scholar Award (to EPB), and NIH Grant GM119707 (to AKB).

## Additional information

### Competing interests

Amie K Boal: Reviewing editor, eLife. The other authors declare that no competing interests exist.

### Funding

| Funder | Grant reference number | Author |
| --- | --- | --- |
| National Science Foundation | 1454007 | Emily P Balskus |
| National Science Foundation | 2003436 | Emily P Balskus |
| Research Corporation for Science Advancement | Cottrell Scholar Award | Emily P Balskus |
| National Institutes of Health | GM119707 | Amie K Boal |

The funders had no role in study design, data collection and interpretation, or the decision to submit the work for publication.

### Author contributions

Nathaniel R Braffman, Terry B Ruskoski, Conceptualization, Formal analysis, Investigation, Writing – original draft, Writing – review and editing; Katherine M Davis, Nathaniel R Glasser, Cassidy Johnson, Investigation; C Denise Okafor, Formal analysis, Investigation, Writing – original draft; Amie K Boal, Conceptualization, Formal analysis, Funding acquisition, Writing – original draft, Writing – review and editing; Emily P Balskus, Conceptualization, Funding acquisition, Writing – original draft, Writing – review and editing

### Author ORCIDs

Nathaniel R Braffman ⓘ http://orcid.org/0000-0001-6563-984X
Amie K Boal ⓘ http://orcid.org/0000-0002-1234-8472
Emily P Balskus ⓘ http://orcid.org/0000-0001-5985-5714

### Decision letter and Author response

Decision letter https://doi.org/10.7554/eLife.75761.sa1
Author response https://doi.org/10.7554/eLife.75761.sa2

## Additional files

### Supplementary files

• Supplementary file 1. Catalog of 715 protein sequences with homology to CylK and/or BrtB from the IMG-JGI database.

• Supplementary file 2. Annotated table of putative CylK homologs that contain N-terminal fusions or are encoded by organisms with CylC- or CylI-like enzymes.

• Transparent reporting form

### Data availability

Diffraction data have been deposited in the PDB under the accession codes 7RON, 7ROO. All other data generated or analyzed during this study and included in the manuscript and supporting files; Source Data files have been provided for Figure 4.

The following datasets were generated:

| Author(s) | Year | Dataset title | Dataset URL | Database and Identifier |
|---|---|---|---|---|
| Ruskoski TB, Boal AK | 2022 | Crystal structure of the Friedel-Crafts alkylating enzyme CylK from Cylindospermum licheniforme | https://www.rcsb.org/structure/7RON | RCSB Protein Data Bank, 7RON |
| Ruskoski TB, Boal AK | 2022 | Crystal structure of Friedel-Crafts alkylating enzyme CylK from Cylindospermum licheniforme with bromide | https://www.rcsb.org/structure/7ROO | RCSB Protein Data Bank, 7ROO |

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
