## [Editor Report]

This work has revealed an unexpected mechanism by which enzyme-catalyzed alkylation occurs. The results presented here will be broadly relevant to those pursuing enzyme engineering as well as efforts aimed toward developing small molecule inhibitors of this unusual enzyme transformation.

---

## [Decision Letter]

**Decision letter after peer review:**

Thank you for submitting your article "Structural Basis for an Unprecedented Enzymatic Alkylation in Cylindrocyclophane Biosynthesis" for consideration by *eLife*. Your article has been reviewed by 2 peer reviewers, and the evaluation has been overseen by a Reviewing Editor and Senior Editor. The following individual involved in review of your submission has agreed to reveal their identity: Aimin Liuof UTSA (Reviewer #2).

The reviewers have discussed their reviews with one another, and the Reviewing Editor has drafted this to help you prepare a revised submission. In general, we were highly enthusiastic about this study and believe the CylK structural and mechanistic insights reported here represent important advances.

Essential revisions:

We agreed that the gain of function mutations/ expanding substrates are not necessary for the current manuscript.

Please address the following issues by Reviewer 2

In abstract:

Unlike Asp440, Glu374 has much less experimental support for its role as a resorcinol nucleophile activating residue. It is not included in the structure-guided mutant screen (Figure 3C), and E374A shows appropriative activity towards the native substrate (Figure 4D). Bioinformatics studies also show that it is not a highly conserved amino acid. It plays a role in facilitating substrate binding, but it is non-essential. Thus, it might be premature to include Glu374 in the abstract and proposed mechanism (Figure 5C).

In Page 3 (bottom): Missing reference for the "initial biochemical studies…"

In the caption of Figure 2, (C) and (D) should be switched. Panel D has cavity 1 and cavity 2 labels. But these labels are not used in the main text, especially in pp. 8 – 9.

In page 8, line 4, "In prokaryotes and plants, …": since plants belong to eukaryotes, it needs to be clarified.

Figure 4C is redundant with Figure S7A. This is a biochemically reasonable assumption that depicts the discussion on page 20 (but not called to either of it). The flip-over requires energy and significant active site conformational changes to break all the existing protein-substrate interactions and turn the half-processed substrates around. I would suggest removing Figure 4C. Also, Figure S7A should be called upon in the discussion (page 20).

In Table S1, the overall CC1/2 value for Native is missing.

In both Table S1 and S4, low-resolution values are missing.

In both Table S1 and S4, 'B-factors' should be 'B-factors (Å2)'.

In both Table S1 and S4, Ramachandran allowed (%) and outlier (%) should be reported. Listing all the outliers in the footnote would be helpful.

Since both are high-resolution structures, I wonder whether B-factor refinement such as TLS carried out accordingly.

*Reviewer #1 (Recommendations for the authors):*

Specifically, to get a thorough understanding of its high regioselectivity and stereospecificity, it is better to expand its substrate scope and verify enzyme's regioselectivity and stereospecificity.

Nearly all mutants show decreased catalytic activity, it is better to find the key residues to increase its reactivity.

*Reviewer #2 (Recommendations for the authors):*

The use of Br- is a brilliant idea in this work. However, it may be strengthened if inorganic anions, including Br-, are shown to be competitive inhibitors. It may suffice if this has already been done (such as certain buffers are inhibitive).

Unlike Asp440, Glu374 has much less experimental support for its role as a resorcinol nucleophile activating residue. It is not included in the structure-guided mutant screen (Figure 3C), and E374A shows appropriative activity towards the native substrate (Figure 4D). Bioinformatics studies also show that it is not a highly conserved amino acid. It plays a role in facilitating substrate binding, but it is non-essential. Thus, it might be premature to include Glu374 in the abstract and proposed mechanism (Figure 5C).

In Page 3 (bottom): Missing reference for the "initial biochemical studies…"

In the caption of Figure 2, (C) and (D) should be switched. Panel D has cavity 1 and cavity 2 labels. But these labels are not used in the main text, especially in pp. 8 – 9.

In page 8, line 4, "In prokaryotes and plants, …": since plants belong to eukaryotes, it needs to be clarified.

Figure 4C is redundant with Figure S7A. This is a biochemically reasonable assumption that depicts the discussion on page 20 (but not called to either of it). The flip-over requires energy and significant active site conformational changes to break all the existing protein-substrate interactions and turn the half-processed substrates around. I would suggest removing Figure 4C. Also, Figure S7A should be called upon in the discussion (page 20).

In Table S1, the overall CC1/2 value for Native is missing.

In both Table S1 and S4, low-resolution values are missing.

In both Table S1 and S4, 'B-factors' should be 'B-factors (Å2)'.

In both Table S1 and S4, Ramachandran allowed (%) and outlier (%) should be reported. Listing all the outliers in the footnote would be helpful.

Since both are high-resolution structures, I wonder whether B-factor refinement such as TLS carried out accordingly.

---

## [Author Response]

Essential revisions:We agreed that the gain of function mutations/ expanding substrates are not necessary for the current manuscript.Please address the following issues by Reviewer 2In abstract:Unlike Asp440, Glu374 has much less experimental support for its role as a resorcinol nucleophile activating residue. It is not included in the structure-guided mutant screen (Figure 3C), and E374A shows appropriative activity towards the native substrate (Figure 4D). Bioinformatics studies also show that it is not a highly conserved amino acid. It plays a role in facilitating substrate binding, but it is non-essential. Thus, it might be premature to include Glu374 in the abstract and proposed mechanism (Figure 5C).

Based on our molecular dynamics work, we speculate that Glu374 primarily plays a role in resorcinol nucleophile binding, and considering the low but still appreciable level activity of the Glu374 to Ala mutant, we agree that Glu374 is likely not the key nucleophile activating residue. To address this comment, we have amended the abstract to only mention Asp440 as the nucleophile activating residue, we have clarified the proposed mechanism (Figure 5C) to indicate that Asp440 likely deprotonates the resorcinol nucleophile while Glu374 may participate in substrate binding, and we have also substituted the word ‘activating’ for ‘interacting’ in the molecular dynamics Results section .

In Page 3 (bottom): Missing reference for the "initial biochemical studies…"

Citation added.

In the caption of Figure 2, (C) and (D) should be switched. Panel D has cavity 1 and cavity 2 labels. But these labels are not used in the main text, especially in pp. 8 – 9.

We have switched captions 2(C) and 2(D), and modified the main text to include these labels.

In page 8, line 4, "In prokaryotes and plants, …": since plants belong to eukaryotes, it needs to be clarified.

We have clarified this statement to say:

“In prokaryotes and some eukaryotes (plants), …”

Figure 4C is redundant with Figure S7A. This is a biochemically reasonable assumption that depicts the discussion on page 20 (but not called to either of it). The flip-over requires energy and significant active site conformational changes to break all the existing protein-substrate interactions and turn the half-processed substrates around. I would suggest removing Figure 4C. Also, Figure S7A should be called upon in the discussion (page 20).

We have removed panel A from Figure S7 because of the redundancies with Figure 4C. However, we would prefer to keep Figure 4C because we think the chemical structures are useful to understand panels 4D and 4E, and the substrate flip-over is helpful to illustrate. To further clarify our proposal and the link to Figure 4C, we have modified the text at the top of page 13. In response to this comment we have also added additional explanatory text in the Discussion section.

“Having discovered that the CylK active site cleft has two potential alkyl halide binding/activating regions, we wondered if the two discrete native alkylation reactions catalyzed by CylK occurred at one or both bromide binding sites. If only a single bromide site is catalytically active, we speculate that the mono-alkylated intermediate 2 is likely released, rotates 180°, and must rebind the active site for the second alkylation event to occur (Figure 4C). To test this proposal, we individually mutagenized several residues closely associated with or appearing to directly bind either bromide, and examined their ability to perform the first and second alkylation reactions with native substrates 1 and 2.”

“Finally, intermediate 2 is released from the active site, must reorient 180°, and re-bind for the second intramolecular alkylation to occur in the same fashion. This reorientation likely requires significant energy to disrupt protein-substrate interactions and might suggest active site conformational changes occur to accommodate intermediate 2 in the catalytically competent orientation, however, this warrants further investigation.”

In Table S1, the overall CC1/2 value for Native is missing.

We have now included the CC1/2 value in Table S1 (now Table 1).

In both Table S1 and S4, low-resolution values are missing.

Low resolution values are now included in both tables (now Table 1 and 5).

In both Table S1 and S4, 'B-factors' should be 'B-factors (Å2)'.

Units were added to the B-factor entry in both tables (now Table 1 and 5).

In both Table S1 and S4, Ramachandran allowed (%) and outlier (%) should be reported. Listing all the outliers in the footnote would be helpful.

Ramachandran allowed (%) and outlier (%) were added to Table S1 and S4 (now Table 1 and 5). A description of the single outlier observed can be found in the methods section.

Since both are high-resolution structures, I wonder whether B-factor refinement such as TLS carried out accordingly.

We thank the reviewer for this suggestion. We performed TLS refinement with both datasets, and one was improved by this approach. We have now updated Table S1 (now Table 1) and our coordinates in the PDB.

Reviewer #1 (Recommendations for the authors):Specifically, to get a thorough understanding of its high regioselectivity and stereospecificity, it is better to expand its substrate scope and verify enzyme's regioselectivity and stereospecificity.Nearly all mutants show decreased catalytic activity, it is better to find the key residues to increase its reactivity.

Though we eventually aim to discover CylK mutants with an expanded substrate scope, we feel that this work is beyond the scope of this current manuscript.

Reviewer #2 (Recommendations for the authors):The use of Br- is a brilliant idea in this work. However, it may be strengthened if inorganic anions, including Br-, are shown to be competitive inhibitors. It may suffice if this has already been done (such as certain buffers are inhibitive).

The potential for bromide, chloride, or other inorganic anions to be inhibitory is interesting, especially considering the relatively high levels of chloride in our buffers. We have found certain levels of DMSO and glycerol in assay mixtures to be somewhat inhibitory, but we believe determining the mode of inhibition for these and other inhibitors is beyond the scope of this work.